# Seeds of Structure: Patch PCA Reveals Universal Compositional Cues in Diffusion Models

**Qingsong Wang**
Halıcıoğlu Data Science Institute
University of California, San Diego
La Jolla, CA 92093
qswang92@gmail.com

**Zhengchao Wan**
Department of Mathematics
University of Missouri
Columbia, MO 65211
zwan@missouri.edu

**Mikhail Belkin**
Halıcıoğlu Data Science Institute
University of California, San Diego
La Jolla, CA 92093
mbelkin@ucsd.edu

**Yusu Wang**
Halıcıoğlu Data Science Institute
University of California, San Diego
La Jolla, CA 92093
yusuwang@ucsd.edu

## Abstract

Diffusion models transform random noise into images of remarkable fidelity, yet the structure of this noise-to-image map remains largely unexplored. We investigate this relationship using patch-wise Principal Component Analysis (PCA) and empirically demonstrate that low-frequency components of the initial noise predominantly influence the compositional structure of generated images. Our analyses reveal that noise seeds inherently contain universal compositional cues, evident when identical seeds produce images with similar structural attributes across different datasets and model architectures. Leveraging these insights, we develop and theoretically justify a simple yet effective Patch PCA denoiser that extracts underlying structure from noise using only generic natural image statistics. The robustness of these structural cues is observed to persist across both pixel-space models and latent diffusion models, highlighting their fundamental nature. Finally, we introduce a zero-shot editing method that enables injecting compositional control over generated images, providing an intuitive approach to guided generation without requiring model fine-tuning or additional training.

## 1 Introduction

Diffusion models [27, 13, 29, 11] have emerged as a powerful class of generative models, achieving remarkable success in image synthesis [26, 9] and more [32, 14]. These models operate by gradually transforming random noise into structured data through an iterative denoising process. Within the image generation domain—our focus in this work—the noise-to-image map exhibits striking consistency across different architectures and training procedures [37], suggesting intrinsic patterns underlying this transformation that transcend specific implementation details. Understanding these patterns, even approximately, has significant practical value. Recent work [23, 19, 39, 36] has shown that choices of initial noise seeds significantly influence the alignment between generated images and text prompts, particularly regarding global structure and composition. However, precisely how specific noise patterns influence particular image characteristics remains an open question with substantial implications for controlled generation and image editing.

In this paper, we investigate the influence of initial noise on generated image structure through a patch-wise Principal Component Analysis (PCA) framework [25, 7]. Using this approach, we empirically

39th Conference on Neural Information Processing Systems (NeurIPS 2025).

demonstrate that the eigenspace identified by patch principal components positively correlates with principal variation directions of the noise-to-image map. We focus on the low-frequency components identified by patch PCA, which in the image patch domain correspond to illumination and major structural lines, contributing to the perceived composition of an image.

Notably, when the low-frequency components of the initial noise are fixed and only high-frequency components are perturbed, the generated images exhibit minimal structural changes. This suggests that low-frequency noise components predominantly determine the compositional structure in diffusion generation. Additionally, our empirical analyses reveal that noise seeds inherently contain *universal compositional cues*, so images generated from identical noise seeds across different datasets exhibit similar compositional attributes, evidenced through visual examples and quantitative metrics. This finding helps explain phenomena such as the observed variability in text-to-image generation when aligning prompts with global structure and composition across different noise seeds; the seeds themselves, while random, carry distinct instantiations of these underlying compositional templates.

**Proposed Methods.** The robustness of these compositional cues enables us to develop a simple yet effective *Patch PCA denoiser* that extracts underlying structure from noise using only a generic set of natural images. This approach is theoretically justified by Theorem 5.1, where we prove that the Patch PCA denoiser is optimal for the diffusion model loss when the denoiser calculates each pixel's output based solely on local patch—common in convolutional architectures—and the patch distribution is Gaussian, a reasonable assumption for small natural image patches [40]. Despite its simplicity, images generated by our Patch PCA denoiser exhibit structural similarities to those from sophisticated neural networks, though less visually refined.

Structural similarities become increasingly apparent when projecting onto leading PCA subspaces, demonstrating these compositional cues primarily reside in low-frequency components. These findings extend to latent diffusion models like Stable Diffusion [26], where the same structural hierarchy persists despite operating in a compressed latent space. This cross-model consistency reveals noise seeds contain intrinsic compositional cues that transcend specific architectures, representing fundamental properties of the generative process itself.

Leveraging these insights, we develop a zero-shot *Patch PCA based noise editing* method that selectively interpolates noise representations with reference images within specific PCA frequency bands. Our approach enables structural control without model fine-tuning or additional training, providing an efficient method for guided image generation by directly manipulating the compositional blueprint in the noise.

## 2  Related Work

**Noise Seeds and Composition in Diffusion Models.** Recent studies show that initial noise seeds significantly influence compositional structure in diffusion-generated images, with certain seeds consistently producing better-aligned results for specific prompts [23, 36]. This has led to optimization methods for finding high-performing seeds [39, 19]. While these works establish the importance of noise seeds, the underlying mechanisms of how seeds encode compositional information remain an open area for exploration. Our work contributes to this understanding through patch PCA analysis, demonstrating that compositional cues predominantly reside in low-frequency noise components, propose a zero-shot editing method that can guide the composition of generated images.

**Inductive Bias in Diffusion Models.** Recent works on diffusion models have explored various aspects of inductive bias. Studies like [20, 34] observe linearity in well-trained diffusion models and develop closed-form approximations under Gaussian assumptions, while [24, 15] examine these models through local image patches, arguing that convolutional architectures constrain each pixel's output to depend only on its surrounding patch. Our work builds upon these insights but takes a different direction—instead of seeking optimal theoretical surrogates for specific datasets, we aim to identify universal compositional structures that generalize across diverse data distributions. By leveraging the observation that local patches follow Gaussian distributions more accurately than full images [40], we extract fundamental compositional cues shared across different domains without dataset-specific optimization. This approach allows us to analyze and manipulate compositional elements embedded in the noise-to-image map that persist across varied model architectures, enabling practical applications like zero-shot noise editing without requiring dataset-specific tuning or training.

# 3 Preliminaries

**Diffusion Models and Denoisers.** Diffusion models generate data by reversing a gradual noising process. When parameterized by noise-to-signal ratio $\sigma$ as in Song et al. [28], the reverse process follows the ODE:

$$d\mathbf{x}_\sigma/d\sigma = (D(\mathbf{x}_\sigma, \sigma) - \mathbf{x}_\sigma)/\sigma \tag{1}$$

where $p_\sigma = p * \mathcal{N}(\mathbf{0}, \sigma^2 \mathbf{I})$ is the data distribution convolved with Gaussian noise and $D(\mathbf{x}_\sigma, \sigma)$ is the denoiser. A neural network is typically trained to approximate the denoiser:

$$\mathbb{E}_{\sigma \in (0,\infty), \mathbf{x}_\sigma \sim p_\sigma} \|D(\mathbf{x}_\sigma, \sigma) - \boldsymbol{x}\|^2, \tag{2}$$

and sampling is performed by integrating Equation (1) starting from a random noise.

**PCA and linear optimal denoisers.** For a data distribution $p(\mathbf{x})$ in $\mathbb{R}^d$ with mean $\boldsymbol{\mu}$ and bounded covariance $\boldsymbol{\Sigma}$, we can decompose the covariance matrix $\boldsymbol{\Sigma}$ into its eigenvalues and eigenvectors, $\boldsymbol{\Sigma} = \sum_{i=1}^d \lambda_i \mathbf{u}_i \mathbf{u}_i^T$, where $\lambda_i$ are the eigenvalues and $\mathbf{u}_i$ are the corresponding eigenvectors. This is the principal component analysis (PCA) of the data distribution and the eigenvectors are called the principal components. It is shown in [20, Theorem 1] that the optimal linear approximation of the denoiser at noise level $\sigma$ is given by:

$$D_{\mathrm{PCA}}(\mathbf{x}_\sigma, \sigma) = \boldsymbol{\mu} + \sum_{i=1}^d \frac{\lambda_i}{\lambda_i + \sigma^2} \langle \mathbf{x}_\sigma - \boldsymbol{\mu}, \mathbf{u}_i \rangle \mathbf{u}_i, \tag{3}$$

which performs scaling along the principal components of the data distribution.

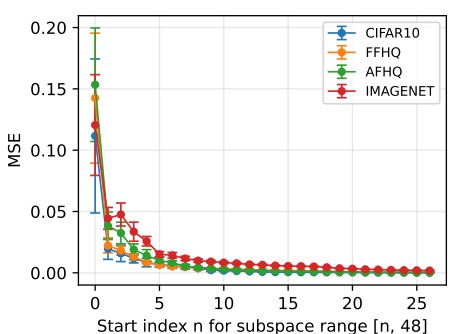

(a) MSE across frequency bands perturbations.

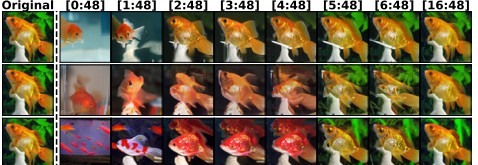

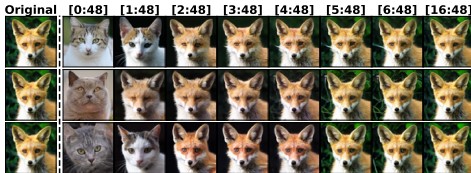

(b) ImageNet (top), AFHQ (bottom)

Figure 1: **Frequency band perturbation analysis.** (a) MSE between original and perturbed-noise generated images across frequency bands. When perturbing including low-frequency bands, the MSE is significantly higher than when only high-frequency bands are perturbed. (b) Visual examples from ImageNet (top) and AFHQ (bottom): original image followed by perturbations with full frequency bands $[n, 48]$ where $n = 0, 1, 2, \cdots, 6, 16$. Three different perturbations are shown for each frequency band.

**ODE sampler and the noise-to-image flow map.** The ODE sampler [28], popular for its simplicity and effectiveness, is deterministic. For a convergent Ordinary Differential Equation (ODE), the final image is uniquely determined by the initial noise. This relationship defines a mapping, known as a *flow map*, from the noise space to the data manifold. The convergence of the ODEs in diffusion models, and the existence of a flow map, have been established in recent studies [12, 33]. The flow map imposes a structure on the noise space. For example, for a simple dataset of two points $\{-1, 1\} \subset \mathbb{R}$, the flow map partitions the noise space $\mathbb{R}$ into two regions, with all noises from the negative axis mapped to $-1$ and all noises from the positive axis mapped to $1$. Hence, the final sampled points are fully determined by the sign of the initial noise. It is of great interest to find a similar structure in the noise space of image diffusion models.

# 4 Probing the Noise–to–Image Map through Patch PCA filter

In this section, we probe the noise-to-image map in pre-trained diffusion models by analyzing how perturbation in *initial noise* influences the generated images. Our diagnostic tool is Patch PCA, which effectively captures image statistics and has been shown to be valuable for image denoising tasks [2].

**Methodology:** We first compute a generic patch PCA covariance matrix using $4 \times 4$ patches extracted from 10,000 randomly sampled images from the ImageNet dataset and compute the covariance matrix with its eigen-decomposition. In fact, for small patches (size less than $16 \times 16$), the covariance matrices computed from differently collected images are almost identical; see Figure 9 in the Appendix for an ablation study. With 3 color channels, each image patch is a $4 \times 4 \times 3$ tensor, and we obtain 48 eigenvalue-eigenvector pairs from the eigen-decomposition. As expected, the eigenvalues follow a characteristic power-law distribution typical of natural images, with the leading eigenvectors representing color/illumination variations, followed by edge detectors and then higher frequency components capturing refined textures (see Figure 10a and Figure 10b in the Appendix). We conduct experiments using pre-trained EDM models [17] on diverse datasets: ImageNet [8] ($64 \times 64$), FFHQ, AFHQ, and CIFAR-10 [18]. We fix 128 random noise initializations and generate reference images using the pre-trained models. For each noise initialization, and for each frequency band $[n, 48]$ (where $n$ ranges from 0 to 48), we generate 100 perturbed variants by subdividing the noise tensor into non-overlapping $4 \times 4$ patches and resampling the identified frequency bands $[n, 48]$ on each patch; see Algorithm 2 for the details. This ensures independence and preserves the standard Gaussian distribution due to the orthogonality of the PCA basis. We then measure the mean squared error (MSE) between images generated from the perturbed noise and the original reference images.

**Results:** The analysis in Figure 1a reveals a clear pattern: perturbations in low-frequency components (small $n$) induce substantial changes in the generated images, while high-frequency perturbations (large $n$) have minimal impact. Specifically, when $n$ exceeds 20, the MSE remains consistently low (below 0.01, with pixel values in $[0, 1]$), and the generated images in Section 3 are visually indistinguishable from the originals. This finding is indicative that the noise-to-image map in diffusion models preserves a hierarchical structure aligned with the patch PCA decomposition: low-frequency components control fundamental aspects such as illumination and global layout, while high-frequency components primarily contribute to texture and fine details.

**Cross-dataset compositional resemblance:** We observe that images generated from identical noise seeds across different datasets exhibit similar structural patterns. In Figure 1b, goldfish (ImageNet) and fox (AFHQ) generated from the same seed share similar lighting patterns, with matching bottom-left highlighting and bottom-right edge shapes. This compositional correspondence persists across datasets (see Figure 2a, first three rows). To quantitatively validate this visual similarity, we compute both MSE and structural similarity index measure (SSIM) [35] between pairs of images generated from identical noise seeds **across** different datasets. SSIM is particularly valuable for this analysis because, unlike MSE which measures pixel-level differences, SSIM captures structural similarities by considering luminance, contrast, and structure; see Appendix A.2 for the detailed definition. This makes SSIM especially suited for detecting compositional resemblances that may not be apparent in pure pixel-wise comparisons. As shown in Table 1, both MSE and SSIM scores show significant closeness for seed-paired images (SSIM: 0.423-0.469; MSE: 0.032-0.041) compared to randomly paired images (SSIM: 0.054-0.065; MSE: 0.128-0.136). This confirms that the generated images share structural properties when derived from the same noise initialization across different datasets, providing strong evidence that the initial noise contains universal compositional cues that transcend dataset boundaries and training domains. We include ablation studies on ODE sampling schedules, network architectures (U-Net versus transformer-based), and flow-matching variants in Appendix A.2, where we consistently observe the same seed-dependent structural pattern.

# 5 Uncovering Universal Compositional Cues Through Patch PCA Denoiser

Our cross-dataset experiments revealed consistent compositional patterns when using identical noise initializations. This suggests that initial noise may contain universal compositional cues that transcend specific datasets. We now formalize and validate this hypothesis by directly extracting these compositional elements using our patch PCA framework.

Table 1: **Image Similarity Between Networks and Patch PCA.** SSIM and MSE are computed between images generated from identical noise seed and randomly paired noises across different architectures. Same-seed images show significantly higher similarity (higher SSIM and lower MSE scores) compared to random pairs.

| Network Pair | SSIM (*higher is better*) | | MSE (*lower is better*) | |
|---|---|---|---|---|
| | **Same Seed** | **Random** | **Same Seed** | **Random** |
| ImageNet vs FFHQ | $0.423 \pm 0.087$ | $0.065 \pm 0.040$ | $0.041 \pm 0.017$ | $0.136 \pm 0.055$ |
| ImageNet vs AFHQ | $0.447 \pm 0.097$ | $0.062 \pm 0.039$ | $0.038 \pm 0.018$ | $0.130 \pm 0.055$ |
| FFHQ vs AFHQ | $0.469 \pm 0.074$ | $0.054 \pm 0.041$ | $0.032 \pm 0.012$ | $0.128 \pm 0.046$ |
| Patch PCA vs ImageNet | $0.474 \pm 0.137$ | $0.031 \pm 0.027$ | $0.049 \pm 0.025$ | $0.115 \pm 0.037$ |
| Patch PCA vs FFHQ | $0.473 \pm 0.077$ | $0.029 \pm 0.033$ | $0.045 \pm 0.014$ | $0.114 \pm 0.027$ |
| Patch PCA vs AFHQ | $0.548 \pm 0.084$ | $0.029 \pm 0.034$ | $0.036 \pm 0.014$ | $0.104 \pm 0.029$ |

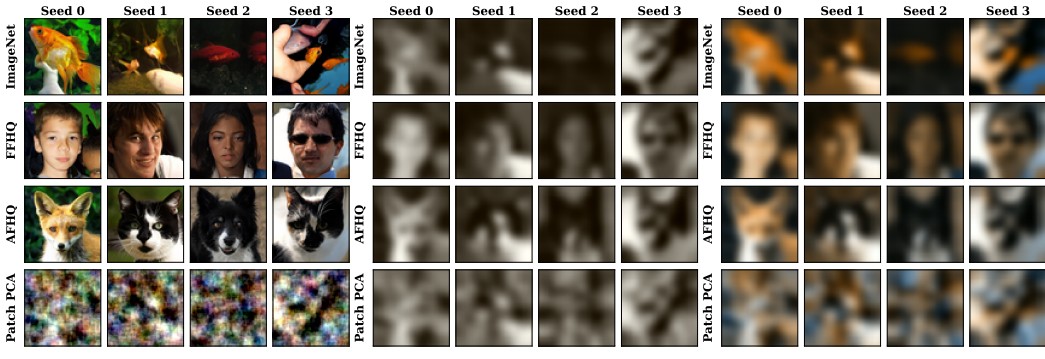

(a) Generated images  (b) Top-1 subspace projection  (c) Top-2 subspace projection

Figure 2: Universal compositional cues across datasets and models. **(a)** Images from identical seeds on ImageNet, FFHQ, AFHQ, and Patch PCA (bottom). **(b)** Top-1 projection revealing consistent illumination patterns. **(c)** Top-2 projection with additional similar color patterns.

**Patch PCA Denoiser.** We propose a simple yet effective Patch PCA denoiser that operates directly on a noisy image $\mathbf{x}_\sigma$, the output of the diffusion model at time step $\sigma$ reads

$$D(\mathbf{x}_\sigma, \sigma) = \mathcal{U}(D_{\mathrm{PCA}}(\mathcal{P}(\mathbf{x}_\sigma), \sigma)). \tag{4}$$

Here $\mathcal{P}$ represents the *patchification operation* that extracts overlapping $p \times p \times c$ where $p$ is the patch size and $c$ is the number of channels, patches with necessary padding from the input image $\mathbf{x}_\sigma$ centered on each pixel. $D_{\mathrm{PCA}}$ is the linear PCA denoiser applied to each patch $\mathbf{p}_i$ in $\mathcal{P}(\mathbf{x}_\sigma)$ via $D_{\mathrm{PCA}}(\mathbf{p}_i, \sigma) := \boldsymbol{\mu} + \sum_{j=1}^{p^2 c} \frac{\lambda_j}{\lambda_j + \sigma^2} \langle \mathbf{p}_i - \boldsymbol{\mu}, \mathbf{u}_j \rangle \mathbf{u}_j$, where $\boldsymbol{\mu}$ is the mean patch and $\{\mathbf{u}_j, \lambda_j\}$ are the eigenvectors and eigenvalues of the patch covariance matrix computed from a generic set of (clean) images and fixed throughout the experiments. $\mathcal{U}$ is the reconstruction operation that uses the center pixel of each patch to reconstruct the image. This linear approximation utilizes the idea that compared with the whole image, the patches are closer to being Gaussian distributed, where the linear denoiser is optimal. The patch PCA denoiser can be efficiently implemented using a convolutional layer with a $p \times p \times c$ kernel. In our experiments, we found the approach robust to different patch sizes ranging from 5 to 31, with samples generated from the same noise seed showing consistent results in terms of various metrics; see Appendix A.3 for detailed ablation studies. We use the Patch PCA denoiser in the DDIM sampling [28] (see Equation (1)) to generate images.

Note that convolutional backbones such as CNNs or U-Nets constrain each output pixel to depend only on a fixed-size local patch. Building on the analytic framework of [15], we show that—under a Gaussian patch prior—the best denoiser within this "patch-limited" class is nothing but exactly the patch PCA denoiser; see Theorem A.2 in the Appendix for a more formal statement and proof.

**Theorem 5.1** (Patch PCA denoiser is optimal (Informal)). *Assume that the patch distribution follows a Gaussian law with mean $\mu$ and covariance $\Sigma$, and a candidate denoiser must compute every pixel from its own patch only. Then the minimizer of the standard diffusion (denoising) loss (cf. Equation (2)) over all such candidate denoisers is exactly the patch PCA defined in Equation (4) in most pixels (exluding pixels close to the boundary of an image).*

**Extracting Universal Compositional Cues.** We extract patches from randomly sampled images from the ImageNet and compute the covariance matrix of the patches to construct our Patch PCA denoiser, which in turn is used in DDIM sampling [28] (see Equation (1)) to generate images. The resulting images, shown in the bottom row of Figure 2a, while not as visually refined as neural network outputs, display remarkable compositional similarities when compared with neural network generations from identical noise seeds. The Patch PCA denoised images exhibit fundamental illumination patterns and structural lines that closely match those in their neural network counterparts. This suggests that our simple patch linear approach effectively captures the underlying compositional cues embedded in the noise seeds.

To quantitatively validate that our Patch PCA denoiser captures the same underlying compositional information as sophisticated diffusion models, we perform a systematic comparison of images generated from identical noise initializations. The results, presented in the bottom three rows of Table 1, reveal that Patch PCA denoised images achieve substantially higher SSIM scores with diffusion-generated images when sharing the same noise seed (0.474-0.548) compared to random pairings (0.029-0.031). Notably, these similarity values exceed even those between diffusion models trained on different datasets (0.423-0.469), providing strong statistical evidence that our linear approach successfully extracts universal compositional information encoded in the noise. The MSE also shows significantly lower error between same-seed pairings (0.036-0.049) compared to random pairings (0.104-0.115). Visual inspection in Figure 2a confirms that common illumination gradients, dominant structural lines, and color distribution patterns remain consistent across all images derived from the same seed, regardless of whether they were generated by domain-specific neural networks or our domain-agnostic Patch PCA method. This suggests that our approach effectively extracts the compositional cues that diffusion models inherently follow while neural networks trained on different datasets add domain-specific refinements and details.

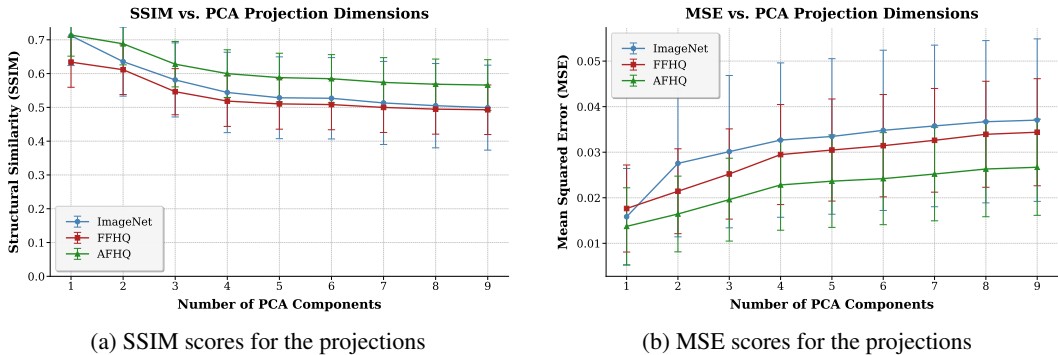

(a) SSIM scores for the projections        (b) MSE scores for the projections

Figure 3: SSIM and MSE scores for the projections of the generated images from the same noise seed across different datasets. The scores improve when the images are projected onto subspaces with leading eigenvalues, indicating that the patch PCA model captures the similar structural elements across datasets that are preserved in low-frequency components.

**Amplifying Compositional Cues Through Subspace Projection.** To further investigate and visualize the compositional information encoded in the noise, we examine the lower frequency projections of the generated images. For each generated image by neural networks or the Patch PCA denoiser, we first extract patches from the image, project the patches onto the leading eigenvectors of the covariance matrix, and then reconstruct the image from patches by folding back the patches.

The visualizations in Figure 2b reveal that restricting to just the first principal component preserves key illumination patterns and major structural lines while eliminating texture and fine details. The compositional resemblance across images from the same noise seed is amplified in this highly restricted representation. When expanding to include the second principal component (Figure 2c), color variations emerge while maintaining coherence across datasets and the Patch PCA model.

We quantify this phenomenon in Figure 3a and Figure 3b, where both SSIM and MSE metrics show systematic improvement as images are compared in these increasingly restricted representations. This progressive enhancement in similarity confirms that the essential compositional information is concentrated in just the first few PCA components, corresponding to the most basic structural elements of the images. These findings complement our earlier perturbation experiments in Section 4,

providing compelling evidence that diffusion models across different datasets follows fundamental compositional structure in the low-dimensional components of noise identified by the patch PCA.

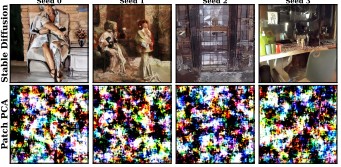 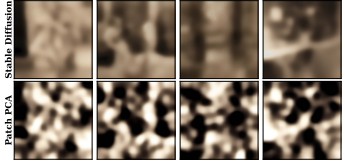 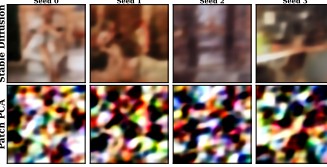

(a) Generated images     (b) Top-1 subspace projection     (c) Top-5 subspace projection

Figure 4: Visualization of the generated images from Stable Diffusion (top row) and Patch PCA model (bottom row) using uncurated seeds 0-3. The first column shows generated images from the same noise seed across different datasets, while the second and third columns show patch PCA denoised images projected onto the top-1 and top-5 subspaces, respectively. The consistent structural similarities demonstrates the effectiveness of our patch PCA denoiser in latent diffusion setting.

**Patch PCA with Latent Diffusion Models** We now examine whether our findings extend to latent diffusion models, which operate in a compressed latent space rather than pixel space. We focus on Stable Diffusion (V1.4) [26], adapting our patch PCA framework to its VAE latent space.

Our adaptation involves extracting $p \times p$ patches from encoded ImageNet samples (shape $[4, p, p]$ from latent representations of dimensions $[4, 64, 64]$), computing the covariance matrix, and performing eigen-decomposition. When decoded back to pixel space, the leading eigenvectors (Figure 11 in Appendix A.3) reveal a striking phenomenon: they mirror the patterns observed in pixel-space models, with initial components corresponding to color/illumination variations, followed by edge detectors at higher frequencies. This cross-domain consistency suggests the VAE decodes the latent space patch eigenvectors into pixel space patch eigenvectors, which could be helpful for preserving structural hierarchies across representation domains and expanding the applicability of our patch PCA approach to various diffusion architectures. This perspective is further validate by the empirical observation in Table 7 in Appendix A.3 where the encoded pixel-patch and latent patch top-$k$ eigenspaces are closely aligned (mean cosine of principal angles $\approx 0.93$ for top 100 subspace).

We apply our Patch PCA denoiser from Equation (4) to Stable Diffusion's latent space and visualize the results in Figure 4 for four uncurated seeds (0-3). The first column shows standard Stable Diffusion generations (with empty prompt) while the second and third columns display patch-based projections onto the top-1 and top-5 subspaces, respectively. Consistent with our pixel-space findings, projections onto the top-1 subspace isolate illumination and major structural lines, while projections onto the top-5 subspace preserve additional color variations and some edge details. The strong visual alignment between generated and patch PCA denoised images confirms that noise seeds encode consistent compositional elements regardless of whether generation occurs in pixel or latent space.

Quantitatively, SSIM scores between Stable Diffusion and patch PCA images are significantly higher for seed-paired (mean SSIM 0.42) comparisons than random pairs (mean SSIM 0.18) in Figure 5a. The SSIM scores increase when projecting onto lower-dimensional spaces, reaching 0.816 for top-1 projection (Figure 5b). Notably, we observe a slight dip at the top-2 projection due to color tone differences between the models, with Stable Diffusion exhibiting a red-brownish tone not present in the patch PCA projections (Figure 13). These findings suggest that similar compositional structures may be encoded in noise seeds across both pixel-space and latent diffusion models, indicating that low-frequency noise components play an important role in determining structural elements in different model architectures.

## 6 Zero-shot Patch PCA based Noise Editing

Our analysis reveals that low-frequency components of initial noise significantly influence the compositional structure of diffusion-generated images. Building on this insight, we propose a zero-shot editing method that directly manipulates noise through patch PCA frequency band filtering, enabling compositional control **without** requiring model fine-tuning or optimization procedures.

We demonstrate our approach by imposing specific compositional structures from a reference image onto generated outputs. We choose CIFAR-10 for our primary experiments as it provides an unbiased

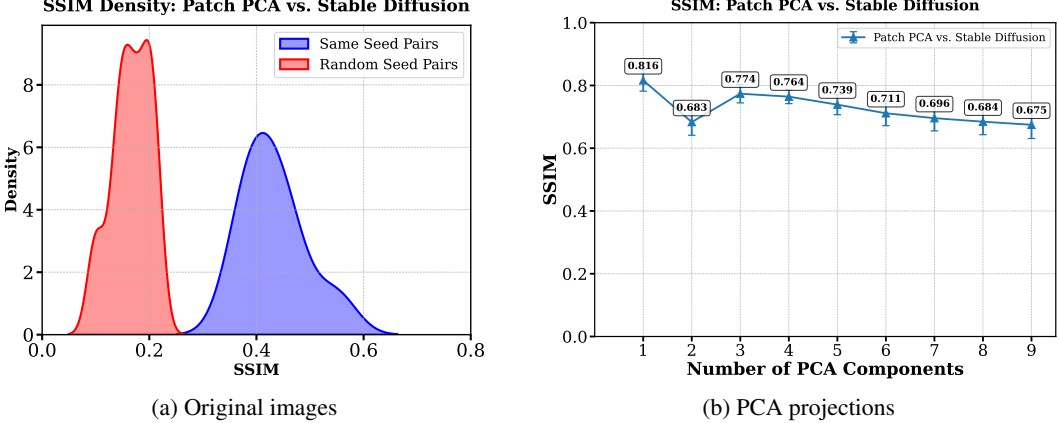

(a) Original images            (b) PCA projections

Figure 5: SSIM score distributions comparing seed-paired vs. randomly-paired images. **(a)** Original images show higher similarity with identical noise seeds. **(b)** PCA projections show increased similarity when projected onto lower-dimensional spaces.

---

**Algorithm 1** Patch-based PCA Noise Editing with Frequency Band Control

---

1: **Input:** Original noise $\mathbf{z}$, reference image $\mathbf{x}_r$, patch size $p$, number of channels $c$, eigenvectors $\{\mathbf{u}_j\}_{j=1}^{c \cdot p^2}$, interpolation factor $\alpha$, frequency indices $\mathcal{I} \subseteq \{1, 2, \ldots, c \cdot p^2\}$
2: **Output:** Edited noise $\mathbf{z}'$
3: Extract overlapping $p \times p$ patches from $\mathbf{z}$ and $\mathbf{x}_r$ to obtain $\{\mathbf{p}_i^{\text{noise}}\}$ and $\{\mathbf{p}_i^{\text{ref}}\}$
4: **for** each patch index $i$ in parallel **do**
5:      Decompose in PCA basis: $\mathbf{p}_i^{\text{noise}} = \sum_{j=1}^{c \cdot p^2} c_{i,j}^{\text{noise}} \mathbf{u}_j$ and $\mathbf{p}_i^{\text{ref}} = \sum_{j=1}^{c \cdot p^2} c_{i,j}^{\text{ref}} \mathbf{u}_j$
6:      // Split the noise patches $\mathbf{p}_i^{\text{noise}}$ into editing part and orthogonal part
7:      $\mathbf{p}_i^{\text{edit}} \leftarrow \sum_{j \in \mathcal{I}} c_{i,j}^{\text{noise}} \mathbf{u}_j$                            ▷ Part to be edited
8:      $\mathbf{p}_i^{\text{orthog}} \leftarrow \sum_{j \notin \mathcal{I}} c_{i,j}^{\text{noise}} \mathbf{u}_j$                         ▷ Preserved part
9:      // Store original norm of the editing part
10:     $\rho_i^{\text{edit}} \leftarrow \|\mathbf{p}_i^{\text{edit}}\|$
11:     // Editing part coefficients
12:     $\mathbf{v}_i^{\text{edit}} \leftarrow \mathbf{0}$
13:     Extract selected coefficients: $\mathbf{c}_i^{\text{noise}} \leftarrow [c_{i,j}^{\text{noise}}]_{j \in \mathcal{I}}$
14:     Extract reference coefficients: $\mathbf{c}_i^{\text{ref}} \leftarrow [c_{i,j}^{\text{ref}}]_{j \in \mathcal{I}}$
15:     Calculate angle between vectors: $\theta \leftarrow \cos^{-1}\left(\frac{\mathbf{c}_i^{\text{noise}} \cdot \mathbf{c}_i^{\text{ref}}}{\|\mathbf{c}_i^{\text{noise}}\| \|\mathbf{c}_i^{\text{ref}}\|}\right)$
16:     Apply SLERP to entire vector: $\mathbf{v}_i \leftarrow \frac{\sin((1-\alpha)\theta)}{\sin(\theta)} \mathbf{c}_i^{\text{noise}} + \frac{\sin(\alpha\theta)}{\sin(\theta)} \mathbf{c}_i^{\text{ref}}$
17:     Reconstruct edited part: $\mathbf{v}_i^{\text{edit}} \leftarrow \sum_{j \in \mathcal{I}} v_{i,j'} \mathbf{u}_j$ where $j'$ is the index of $j$ in $\mathcal{I}$
18:     // Normalize edited part to preserve original energy
19:     $\mathbf{v}_i^{\text{edit}} \leftarrow \rho_i^{\text{edit}} \cdot \frac{\mathbf{v}_i^{\text{edit}}}{\|\mathbf{v}_i^{\text{edit}}\|}$
20:     // Combine edited and orthogonal parts
21:     $\mathbf{p}_i' \leftarrow \mathbf{p}_i^{\text{orthog}} + \mathbf{v}_i^{\text{edit}}$
22: **end for**
23: Reconstruct $\mathbf{z}'$ by using the center pixel of each patch $\mathbf{p}_i'$ to reconstruct the image
24: **Return** $\mathbf{z}'$

---

testing environment where no specific training-guided editing has been performed, allowing us to evaluate the intrinsic capabilities of diffusion models.

**Methodology:** Our patch-based noise editing approach (Algorithm 1) consists of three steps:

1. Identifying specific frequency bands in the patch PCA space that encode compositional information
2. Performing spherical interpolation (SLERP) between the noise and reference patches within these bands
3. Normalizing to preserve the energy of the original noise

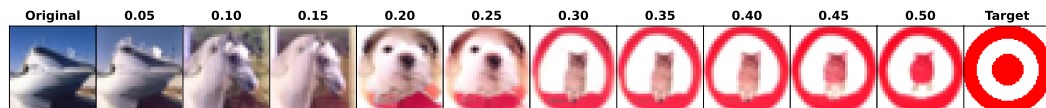

Figure 6: Progression from original image (leftmost, seed 3) to target reference image (rightmost) using zero-shot editing. Middle images are generated with increasing interpolation factors ($\alpha$ from 0.05 to 0.5) using principal components in range $[1, 9]$. Note the gradual emergence of the circular composition from the target image while preserving the semantic content of the original.

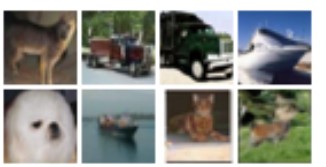
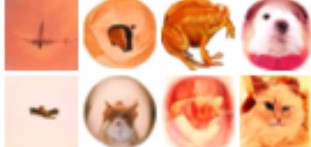
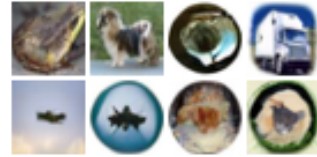

(a) Generations without editing.      (b) Editing in $[1, 9]$ range.      (c) Editing in $[1, 9] \setminus \{2, 5\}$ range.

Figure 7: Zero-shot image editing on Cifar-10. (a) Original images generated from seeds 0-7 without editing. (b) Images generated with noise modified in frequency band $[1, 9]$ using $\alpha = 0.25$. Note the circular composition but with colors influenced by the reference image. (c) Images generated with noise modified in frequency band $[1, 9] \setminus \{2, 5\}$ using $\alpha = 0.25$. By excluding color-encoding eigenvectors, original colors are preserved while circular composition is more prominent.

We utilize the leading 9 eigenvectors from the patch PCA decomposition, which effectively capture essential compositional information in $5 \times 5$ patches. Using a pre-trained EDM model on CIFAR-10, we test our approach with a distinctive red target image (Figure 6, rightmost) as our reference. This reference exhibits both centered and circular compositional properties. We establish random noise initializations (seeds 0-3), with unedited generation results shown in Figure 7a.

**Results and Analysis:** Figure 7 demonstrates the effectiveness of our approach. When editing low-frequency components (Figure 7b), the circular target structure is clearly imposed while maintaining dataset-specific characteristics. Our analysis revealed that eigenvectors 2 and 5 primarily encode color information rather than compositional structure. When these vectors are excluded from the editing process (Figure 7c), the model preserves the original color tones while still adopting the target composition, highlighting the specificity of our patch principal directions. To quantify compositional alignment, we report the SSIM between edited outputs and reference compositions. To assess color preservation independently, we compute the *Bhattacharyya coefficient* [5] (range 0 to 1, higher means more similar) between RGB histograms of the edited image and the original (no-edit) image. A higher Bhattacharyya coefficient indicates better color similarity. These results are summarized in

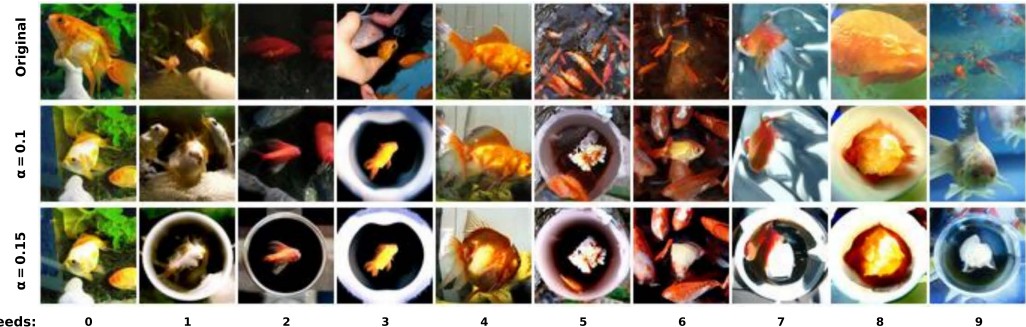

Figure 8: Zero-shot image editing applied to ImageNet goldfish (class 1) using seeds 0-9 with interpolation factors $\alpha = 0.1, 0.15$ and eigenvectors $[1, 9] \setminus \{1, 2, 5\}$. Original images (top) compared to edited versions (middle and bottom). Edited images increasingly adopt the reference circular composition while maintaining semantic content. Note how some seeds (e.g., seed 6) creatively forms the circular pattern while preserving the goldfish identity across interpolation strengths.

Table 2. Together, these metrics confirm that low-frequency editing improves structural alignment with the reference, and by excluding pure color channels eigenvectors, we better preserve the color while maintaining structural control.

Table 2: CIFAR-10, 256 seeds. Low-frequency editing improves structure (SSIM ↑). Channel-aware bands better preserve color.

| Metric | No Edit | Edit $[1, 9]$ | Edit $[1, 9] \setminus \{2, 5\}$ |
|---|---|---|---|
| SSIM vs Ref ↑ | $0.0844 \pm 0.0708$ | $0.2521 \pm 0.0665$ | $0.2887 \pm 0.0812$ |
| Color Hist Corr vs Ref ↓ | $0.0402 \pm 0.0860$ | $0.2515 \pm 0.0827$ | $0.1110 \pm 0.0987$ |
| Color Hist Corr vs No Edit ↑ | — | $0.3809 \pm 0.1137$ | $0.5759 \pm 0.1130$ |

We extend our experiments to ImageNet with class conditioning on "goldfish" (class 1) using the same target-shaped reference. Figure 8 presents uncurated results with seeds 0-9. The results show that our noise manipulation technique successfully guides the model to generate images with the desired circular target composition while maintaining class-specific semantic content. Interestingly, the model exhibits creative variations in implementing the centered circular composition, suggesting an inherent flexibility in how diffusion models interpret and apply compositional constraints. More editing results can be found in Appendix A.4 in the Appendix with different reference images, incorporating masks for localized editing, and its editing effect in the latent space of Stable Diffusion.

Our findings demonstrate a diffusion model's ability to adapt to imposed compositional structures cues while adding dataset-specific details. This suggests that the generative capabilities of these models extend beyond simply reproducing training examples to intelligently combining imposed structural constraints with learned semantic features. The effectiveness of our approach across both CIFAR-10 and ImageNet demonstrates the potential for zero-shot image editing through noise manipulation and the benefit of leveraging frequency bands identified by patch PCA decomposition. We include in the Appendix experiments of noise editing in high-frequency band and object-count alignment settings (see Appendix A.4), which further demonstrate the broad effectiveness of our approach.

# 7 Limitations

While our approach effectively captures compositional cues in diffusion models, some limitations remain. The observed correspondence between pixel-space and latent diffusion model eigenvectors suggests interesting connections that warrant further investigation. A comprehensive evaluation of our training-free zero-shot noise editing in complex multi-object scenes settings is beyond the scope of this work and is left for future exploration.

# 8 Acknowledgments

The authors thank anonymous reviewers for their constructive feedback. This material is based upon work supported by NSF (National Science Foundation) via grants CCF-2112665, DMS-2502083, and DMS-2502084, by the Office of Naval Research (ONR N000142412631), as well as by the Defense Advanced Research Projects Agency (DARPA) under Contract No. HR001125CE020. We also acknowledge the Delta HPC cluster at the NCSA (National Center for Supercomputing Applications), University of Illinois, accessed via the NSF ACCESS program (allocation no. TG-CIS220009).

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

# A Appendix

## A.1 Proof of Theorem 5.1

A distinctive feature of natural images is their spatial coherence, where neighboring pixels exhibit strong correlations. Modern neural architectures like CNNs and vision transformers [10] leverage this property by processing images through local patches. We prove that when a denoiser is constrained to operate locally—computing each pixel's output based solely on its surrounding patch—and when patch statistics follow a Gaussian distribution (a reasonable approximation for small natural image patches [40]), the optimal denoiser is precisely the Patch PCA denoiser defined in Equation (4).

Below, we establish a general result about optimal denoisers within the class of admissible local functions (Definition A.1), extending previous work by [15] on "boundary-broken equivariant local" functions to accommodate arbitrary patch shapes (not necessarily square-shape) and possibly different shapes for each pixel.

**Definition A.1** (Admissible local Functions). We let $D = \{1, \ldots, d\}^2$ denote the pixel lattice and $c \geq 1$ the number of color channels. Let $\Omega_1, \ldots, \Omega_k \subset \mathbb{Z}^2$ be finite patch shapes, each containing the origin as its *center*.

Each pixel $x \in D$ is assigned exactly one patch shape through a *shape–index map* $S : D \to \{1, \ldots, k\}$. For pixel $x$, its assigned patch is defined as $\omega_x := \{(x + r) \mid r \in \Omega_{S(x)}\}$, centered at $x$. Here we assume that the patch assignment satisfies that $\omega_x \subset D$ for all $x \in D$.

A function $f : \mathbb{R}^{d \times d \times c} \to \mathbb{R}^{d \times d \times c}$ is called *admissible local function* if the output $f(I)$ at each pixel $x$ equals the output of a local function $f_{S(x)}$ applied to the surrounding patch $\omega_x(I)$ of the image $I$ at pixel $x$. That is, $f$ is admissible local if there exists, for each shape index $j = 1, \ldots, k$, a measurable map $f_j : \mathbb{R}^{|\Omega_j| \times c} \to \mathbb{R}^c$ such that for every pixel $x \in D$:

$$f(I)(x) = f_{S(x)}(\omega_x(I))$$

where $\omega_x(I)$ is the patch of image $I$ extracted at location $x$ according to shape $\Omega_{S(x)}$.

**Theorem A.2.** *Let $I \in \mathbb{R}^{D \times c}$ be a random image drawn from a distribution $p$ with $\mathbb{E}\|I\|_2^2 < \infty$. For any $\sigma > 0$, we consider the noisy observation defined below*

$$Y = I + \sigma Z, \qquad Z \overset{\text{i.i.d.}}{\sim} \mathcal{N}(0, I), \quad \sigma > 0;$$

*For each patch shape $\Omega_j$, we let $r_j$ denote the law of $\omega_{X_j}(I)$ where $X_j$ is uniformly chosen from $S_j = \{x \in D : S(x) = j\}$. We call $r_j$ the patch distribution of the patch shape $j$. Then, we have that the unique minimizer of the following loss function*

$$\operatorname{argmin}_{f \text{ is admissible local}} \mathbb{E}\big[\|f(Y) - I\|_2^2\big]$$

*can be explicitly determined as follows:*

(i) *For each shape $j = 1, \ldots, k$, we let $Q_j := \omega_{X_j}(I) + \sigma Z_j$ where $Z_j$ is an independent Gaussian noise with the shape $\Omega_j$[1]. Then, we define*

$$f_j^\star(q) = \mathbb{E}\big[(\omega_{X_i}(I))_0 \mid Q_j = q\big], \qquad q \in \mathbb{R}^{|\Omega_j| \times c}, \tag{5}$$

*where $(\omega)_0$ denotes the* central pixel *of patch $\omega$ corresponding to the patch shape $\Omega_i$.*

(ii) *Then, the optimal denoiser $f^\star$ is given by*

$$f^\star(Y)(x) = f_{S(x)}^\star(\omega_x(Y)),$$

*where $f_j^\star$ is defined in equation (5).*

*Proof.* Throughout the proof we use $\mathbb{E}[\cdot]$ for expectation and $\|\cdot\|_2$ for the Frobenius norm on $\mathbb{R}^{d \times d \times c}$. For a patch $\omega$ we write $\omega_0 \in \mathbb{R}^c$ for its *centre pixel* (the entry whose spatial offset is the origin).

For an admissible local function $f$ let

$$R(f) = \mathbb{E}\big[\|f(Y) - I\|_2^2\big] = \sum_{x \in D} \mathbb{E}\big[\|f_{S(x)}(\omega_x(Y)) - I(x)\|^2\big].$$

---

[1]This means that $Z_j$ follows the standard Gaussian distribution on $\mathbb{R}^{|\Omega_j|}$.

Grouping the sum by shape index gives

$$R(f) = \sum_{j=1}^{k} \sum_{x \in S_j} \mathbb{E}\big[\|f_j(\omega_x(Y)) - I(x)\|^2\big], \tag{6}$$

where $S_j = \{x \in D : S(x) = j\}$. Hence each local map $f_j$ influences only the $j$–th summand and can be optimized independently.

Fix a shape index $j$ and choose a pixel $X_j$ uniformly from $X_j$. Set

$$P_j = \omega_{X_j}(I), \qquad Q_j = P_j + \sigma Z_j,$$

where $Z_j$ has i.i.d. $\mathcal{N}(0,1)$ entries indexed by $\Omega_j$.

Note that

$$R_j(f_j) := \sum_{x \in S_j} \mathbb{E}\big[\|f_j(\omega_x(Y)) - I(x)\|^2\big] = |S_j| \cdot \mathbb{E}\big[\|f_j(Q_j) - P_{j,0}\|^2\big]$$

where $P_{j,0}$ is the center pixel of the patch $P_j$. Since $|S_j|$ is a constant factor, minimizing $R_j$ is equivalent to minimizing

$$\widetilde{R}_j(f_j) = \mathbb{E}\big[\|f_j(Q_j) - P_{j,0}\|^2\big], \tag{7}$$

whose unique minimizer is the posterior mean

$$f_j^\star(q) = \mathbb{E}\big[P_{j,0} \,|\, Q_j = q\big], \qquad q \in \mathbb{R}^{|\Omega_j| \times c},$$

which is exactly equation (5). Plugging these local optima into equation (6) gives a global risk $R(f^\star) \le R(f)$ for every admissible local function $f$; if any $f_j$ differs from $f_j^\star$ on a set of positive $q$-measure, the corresponding $\widetilde{R}_j$ (and hence $R$) is strictly larger, proving *uniqueness* of $f^\star$. $\qquad\square$

We now state the rigorous version of Theorem 5.1 below by assuming that the patch prior is Gaussian.

**Corollary A.3** (Closed-form when the patch prior is Gaussian). *Keep the notation of the preceding theorem and assume that, for a fixed patch shape $j$, the patch prior is multivariate normal*

$$r_j = \mathcal{N}\big(\mu_j, \Sigma_j\big), \qquad \mu_j \in \mathbb{R}^{n_j}, \; \Sigma_j \in \mathbb{R}^{n_j \times n_j} \quad n_j := |\Omega_j| \cdot c.$$

*Then, the optimal local map in Equation* (5) *is the following* affine *function*

$$f_j^\star(q) = \mu_{j,0} + [\Sigma_j\big(\Sigma_j + \sigma^2 I_{n_j}\big)^{-1}(q - \mu_j)]_0, \qquad q \in \mathbb{R}^{|\Omega_j| \times c}, \tag{8}$$

*where the subscript* 0 *denotes the* central pixel *of the patch corresponding to the shape $\Omega_j$.*

*In particular, we consider the orthogonal decomposition of $\Sigma_j$ below*

$$\Sigma_j = U_j \Lambda_j U_j^\top, \qquad U_j \in \mathbb{R}^{n_j \times n_j} \text{ orthogonal, } \Lambda_j = \mathrm{diag}(\lambda_{j,1}, \ldots, \lambda_{j,n_j}).$$

*Denote the column eigenvectors by $u_{j,r}$ and the eigenvalues by $\lambda_{j,r}$, $r = 1, \ldots, n_j$. Let $(u_{j,r})_0 \in \mathbb{R}^c$ be the part of $u_{j,r}$ corresponding to the* central pixel *of the patch.*

*Then, Equation* (8) *can be rewritten as follows:*

$$f_j^\star(q) = \mu_{j,0} + \sum_{r=1}^{n_j} \frac{\lambda_{j,r}}{\lambda_{j,r} + \sigma^2} \langle q - \mu_j, u_{j,r} \rangle (u_{j,r})_0, \tag{9}$$

*for every patch $q \in \mathbb{R}^{|\Omega_j| \times c}$.*

*Proof.* Write $P_j \sim \mathcal{N}(\mu_j, \Sigma_j)$ and $Q_j = P_j + \sigma Z_j$ with $Z_j \sim \mathcal{N}(0, I_{n_j})$ independent. The pair $(P_j, Q_j)$ is jointly Gaussian with

$$\mathbb{E}[P_j] = \mu_j, \quad \mathbb{E}[Q_j] = \mu_j, \quad \mathrm{Cov}(P_j, Q_j) = \Sigma_j, \quad \mathrm{Cov}(Q_j) = \Sigma_j + \sigma^2 I_{n_j}.$$

For jointly Gaussian vectors, the conditional mean is affine (see for example [3, Theorem 2.5.1]):

$$\mathbb{E}\big[P_j \,|\, Q_j = q\big] = \mu_j + \Sigma_j(\Sigma_j + \sigma^2 I_{n_j})^{-1}(q - \mu_j).$$

Then, restricting everything to the central pixel gives Equation (8). $\qquad\square$

*Remark* A.4. Now, assume that each pixel is assigned the same patch shape (e.g., a $(2k+1) \times (2k+1)$ square centered at the pixel), with circular padding applied to handle boundary pixels. In this uniform case, we can set $S(x) = 1$ for all $x \in D$, and with the patch prior assumed to be Gaussian, Equation (9) implies that the denoiser for all pixels coincides exactly with the patch PCA denoiser defined in Equation (4). This observation also provides intuition for neural network denoisers: when trained on approximately Gaussian patch distributions and constrained to operate locally, their behavior is close to this optimal Patch PCA denoiser.

## A.2 Supplementary material for Section 4

**Supplementary algorithm for noise perturbation**   We provide the formal algorithm for noise perturbation in Algorithm 2.

---

**Algorithm 2** Patch-PCA Band Resampling for Noise Perturbation

---

1: **Input:** Initial noise $\mathbf{z} \in \mathbb{R}^{64 \times 64 \times 3}$, orthonormal Patch-PCA basis $\{\mathbf{u}_i\}_{i=1}^K$, cutoff index $n$ with $1 \leq n \leq K$, where $K = 3 \times p^2$
2: **Output:** Modified noise $\tilde{\mathbf{z}}$ with resampled high-frequency components
3: Partition $\mathbf{z}$ into disjoint patches $\{\mathbf{z}_i\}_{i=1}^N$ of size $p \times p$ (*we use non-overlapping $4 \times 4$ patches in experiments*).
4: **for** each patch $\mathbf{z}_i$ **do**
5:     Decompose the patch into the Patch-PCA basis:

$$\mathbf{z}_i = \sum_{j=1}^K a_{i,j}\, \mathbf{u}_j, \quad a_{i,j} = \langle \mathbf{z}_i, \mathbf{u}_j \rangle$$

6:     **for** $j = n$ **to** $K$ **do**
7:         Replace coefficient $a_{i,j}$ with a new i.i.d. Gaussian sample:

$$a_{i,j} \leftarrow \epsilon_{i,j}, \quad \epsilon_{i,j} \sim \mathcal{N}(0,1)$$

8:     **end for**
9:     Reconstruct the modified patch:

$$\tilde{\mathbf{z}}_i = \sum_{j=1}^K a_{i,j}\, \mathbf{u}_j$$

10: **end for**
11: Reassemble all patches to obtain the modified noise image $\tilde{\mathbf{z}}$
12: **Note:** Because $\{\mathbf{u}_i\}$ is orthonormal and $\mathbf{z}$ is i.i.d. Gaussian, each $a_{i,j}$ remains Gaussian; therefore, $\tilde{\mathbf{z}} \sim \mathcal{N}(0, I)$.
13: **Return** $\tilde{\mathbf{z}}$

---

**Supplementary on the covariance matrix from the patches**   For small-sized patches, the covariance matrix is robust to the collection of the patches. We sample $10,000$ patches from ImageNet, FFHQ, and AFHQ datasets and extract patches of different sizes, and compute and compare the covariance matrix from the same-sized patches but from different datasets. The result is shown in Figure 9. The covariance matrix is robust to the dataset used for sampling the patches.

**Supplementary similarity index measure (SSIM)**   The SSIM is an alternative to the MSE for measuring the similarity between two images. It is definded in Wang et al. [35] as follows:

$$\text{SSIM}(I, J) = \frac{(2\mu_I \mu_J + C_1)(2\sigma_{IJ} + C_2)}{(\mu_I^2 + \mu_J^2 + C_1)(\sigma_I^2 + \sigma_J^2 + C_2)}, \tag{10}$$

where $I$ and $J$ are the two images, $\mu_I$ and $\mu_J$ are the mean of the images, $\sigma_I^2$ and $\sigma_J^2$ are the variance of the images, $\sigma_{IJ}$ is the covariance between the two images, and $C_1$ and $C_2$ are constants to avoid division by zero. The SSIM is a value between 0 and 1, where 1 means that the two images are identical. The SSIM is a better measure for the structure similarity between two images than the MSE, as it takes into account the luminance, contrast, and structure of the images.

**Supplementary on similarity measure through LPIPS**   The LPIPS is a perceptual similarity measure between two images defined in Zhang et al. [38]. We include the LPIPS results for the

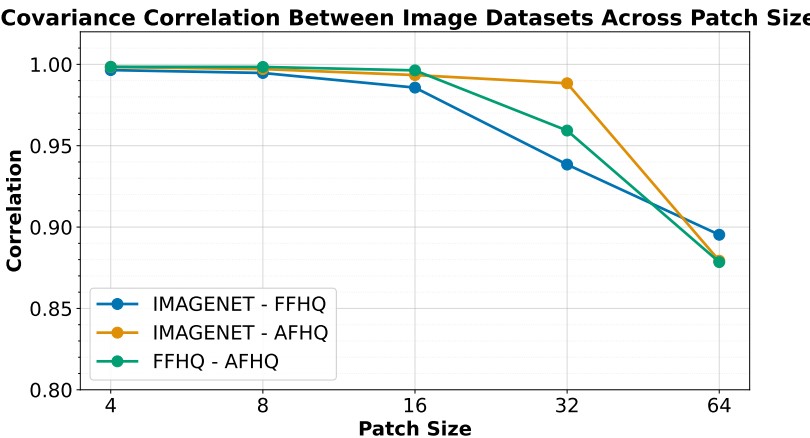

Figure 9: Covariance matrices with different patch sizes constructed from different datasets. For small patch sizes ($\leq 16$), the covariance matrices almost coincide with each other (more than 0.99 correlation).

pixel-space and latent-space models in Table 3 with the same seed and random pairs for both pixel-space and latent-space models. The LPIPS results show a similar trend as SSIM: images generated from identical seeds are consistently closer than those from random pairs. The effect is strongest for pixel-space models such as ImageNet and FFHQ. Comparing with SSIM, LPIPS shows a smaller gap between the same seed and random pairs in general, possibly because LPIPS takes into account the perceptual similarity, which has more emphasis on high frequency components. This matches our central result: seed-dependent information concentrates in low-frequency components, so low-frequency focused similarity metrics like SSIM show a larger effect than high-frequency focused metrics like LPIPS.

Table 3: LPIPS ($\downarrow$) comparison between paired (same-seed) and random pairs across pixel-space and latent-space models. Lower values indicate higher perceptual similarity. Results are averaged over 256 seeds.

| Network Pair | LPIPS (Same Seed) | LPIPS (Random) |
|---|---|---|
| *Pixel-space models* | | |
| ImageNet vs FFHQ | $0.454 \pm 0.069$ | $0.538 \pm 0.064$ |
| ImageNet vs AFHQ | $0.486 \pm 0.099$ | $0.577 \pm 0.093$ |
| ImageNet vs Patch PCA | $0.552 \pm 0.101$ | $0.626 \pm 0.092$ |
| FFHQ vs AFHQ | $0.371 \pm 0.093$ | $0.449 \pm 0.089$ |
| FFHQ vs Patch PCA | $0.542 \pm 0.048$ | $0.584 \pm 0.044$ |
| AFHQ vs Patch PCA | $0.508 \pm 0.067$ | $0.560 \pm 0.063$ |
| *Latent-space models* | | |
| Stable Diffusion vs Patch PCA | $0.895 \pm 0.059$ | $0.951 \pm 0.048$ |

We include additional evidence on the consistency of the seed-dependent structural pattern across different samplers, architectures, and flow-matching variants.

**Deterministic ODE Samplers (EDM, VE, VP)** For a fixed pretrained diffusion model and fixed noise seed $\mathbf{z}$, we compare three common deterministic samplers (default polynomial scheduling in EDM [17], variance exploding (VE) schedule [28], and velocity preserving (VP) schedule [30]). As shown in Table 4, the generated images are nearly identical across samplers, with high SSIM and low MSE between outputs. This indicates that the noise-to-image mapping is robust to the sampler choice in deterministic settings, and our findings are applicable to all of them.

**Network Architectures: U-Net vs Transformer (U-ViT)** To test whether the phenomenon depends on convolutional priors, we compare a U-Net–based EDM model and a transformer-based U-ViT [4] trained on ImageNet-64 using public pre-trained models. Using the same initial noise, we observe

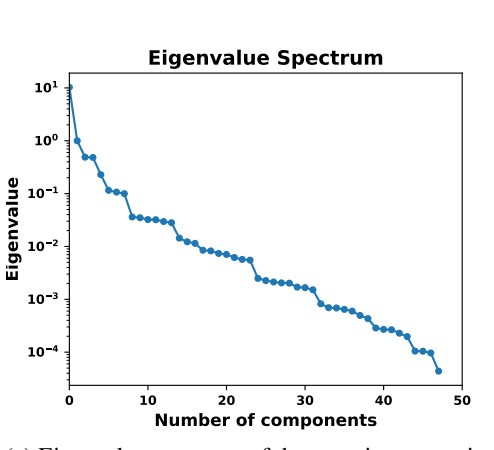

(a) Eigenvalue spectrum of the covariance matrix of $4x4$ patches.

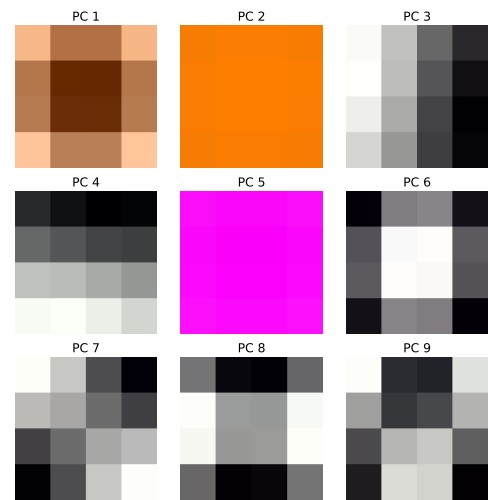

(b) Eigenvectors of the covariance matrix of patches.

Figure 10: Patch PCA spectrum and eigenvectors. The eigenvalues follow a power law distribution, and the leading eigenvectors contain the illumination/color variations, and edge detectors.

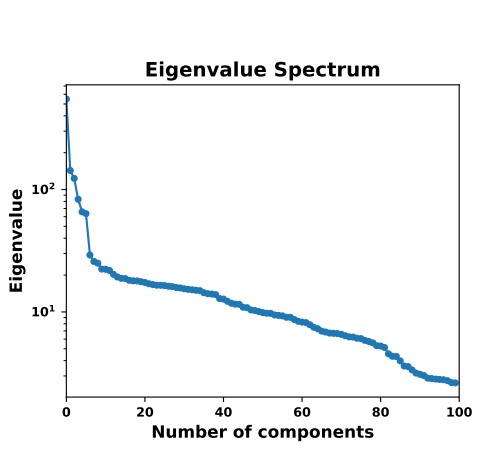

(a) Power spectrum of latent space eigenvectors showing characteristic power-law decay.

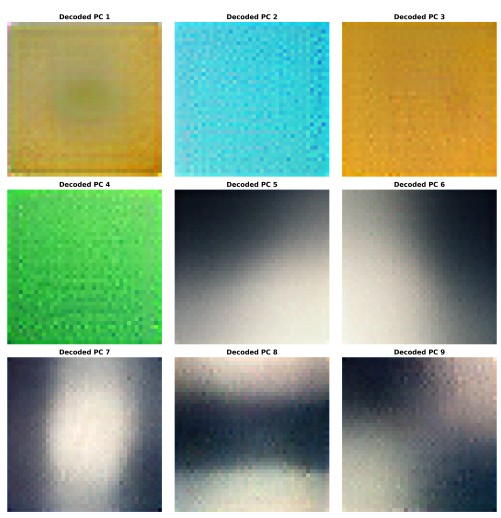

(b) Decoded top latent eigenvectors from $5{\times}5$ patches.

Figure 11: Analysis of latent space PCA: (a) Power spectrum showing characteristic power-law distribution typical of natural image statistics; (b) Visualization of decoded top latent eigenvectors from $5 \times 5$ patches, showing progression from illumination/color variations to edge detectors.

Table 4: Deterministic samplers yield nearly identical outputs for the same seed.

| Sampler Pair | SSIM $\uparrow$ | MSE $\downarrow$ |
|---|---|---|
| EDM vs VE | $0.9902 \pm 0.0275$ | $2.36{\times}10^{-4} \pm 9.76{\times}10^{-4}$ |
| EDM vs VP | $0.9821 \pm 0.0386$ | $5.90{\times}10^{-4} \pm 1.45{\times}10^{-3}$ |
| VP vs VE | $0.9899 \pm 0.0153$ | $2.24{\times}10^{-4} \pm 4.44{\times}10^{-4}$ |

strong structural alignment between generated images. Similar patterns persist when we compare the U-ViT model with our Patch-PCA denoiser.

Table 5: Structure similarity between U-Net and U-ViT architectures using identical seeds.

| Model Pair | SSIM ↑ | MSE ↓ |
|---|---|---|
| U-ViT vs EDM (paired) | $0.8197 \pm 0.0794$ | $0.0065 \pm 0.0045$ |
| U-ViT vs Patch-PCA (paired) | $0.4805 \pm 0.1265$ | $0.0542 \pm 0.0263$ |

The high structural alignment between U-ViT and EDM suggests that the phenomenon of seed-dependent structural pattern is widely applicable to different architectures.

**Flow-Matching variants** The flow-matching (FM) model is a framework that extends the diffusion model with ODE sampler [1, 21, 22]. We train a flow-matching model with the OT scheduling function (or rectified flow) on AFHQ dataset and compare it with the pretrained EDM model on AFHQ dataset. The results are shown in Table 6. We observe that the flow-matching model generates images that are structurally aligned with the EDM model when using the same initial noise. This aligns with the finding in Zhang et al. [37] that the same seed consistently generates similar images across different architectures when trained on the same dataset. Our paper additionally shows that the low-frequency components of generated images from the same seed are aligned across different image datasets.

Table 6: Flow-matching (FM) vs EDM on AFHQ: same-seed generations remain structurally aligned.

| Model Pair | SSIM Paired ↑ | SSIM Random ↑ | MSE Paired ↓ | MSE Random ↓ |
|---|---|---|---|---|
| FM vs EDM | $0.7840 \pm 0.1023$ | $0.0581 \pm 0.0454$ | $0.0104 \pm 0.0068$ | $0.1168 \pm 0.0447$ |
| FM vs PCA | $0.5116 \pm 0.0862$ | $0.0315 \pm 0.0367$ | $0.0394 \pm 0.0149$ | $0.1019 \pm 0.0270$ |

### A.3 Supplementary material for Section 5

**Patch size ablation** We test the effect of different patch sizes on the denoising performance using Equation (4) with different patch sizes. We generate 512 different initial noises and use the different patch sized PCA denoiser as denoiser in the DDIM reverse process Equation (1). We then compare between the generated images from the same initial noise and the different patch sized PCA denoisers in terms of cosine similarity, MSE, and SSIM. The results are shown in Figure 12. We observe that the generated images are visually similar while larger patch sizes produces smoother images as shown in Figure 12a. The cosine similarity, MSE, and SSIM are also similar between the generated images. The quantitative results are shown in Figures 12b to 12d. These results on cosine similarity, MSE, and SSIM show that the PCA denoiser is robust to the patch size used in the denoising.

We also include more projections of the generated images from the latent space PCA denoiser and the Stable Diffusion model for comparison. The projections are shown in Figure 14.

**Pixel-to-Latent Subspace Alignment** Let $V_k = \mathrm{span}\{\mathbf{v}_1, \ldots, \mathbf{v}_k\}$ be the top-$k$ PCA directions of $8p \times 8p$ pixel patches. Let $\mathcal{E}$ be the encoder from Stable Diffusion; define the encoded subspace $E(V_k) = \mathrm{span}\{\mathcal{E}(\mathbf{v}_1), \ldots, \mathcal{E}(\mathbf{v}_k)\}$. Let $W_k$ be the top-$k$ PCA directions of $p \times p$ latent patches. We report the *subspace alignment* score $\mathrm{align}(E(V_k), W_k)$—the mean cosine of principal angles—in Table 7. Pixel-to-latent *subspace alignment* is consistently high (mean cosine $\approx 0.93$–$0.95$), showing that the encoder's dimensionality reduction preserves the leading patchwise PCA directions that our method utilizes.

### A.4 Supplementary material for Section 6

In this section, we provide additional details in Section 6 regarding the zero-shot noise editing precedure.

**Localized editing (though mask)** We apply the noise editing procedure to a specific region of the image by applying a mask to the noise. That is, we only edit the noise in the region defined by the

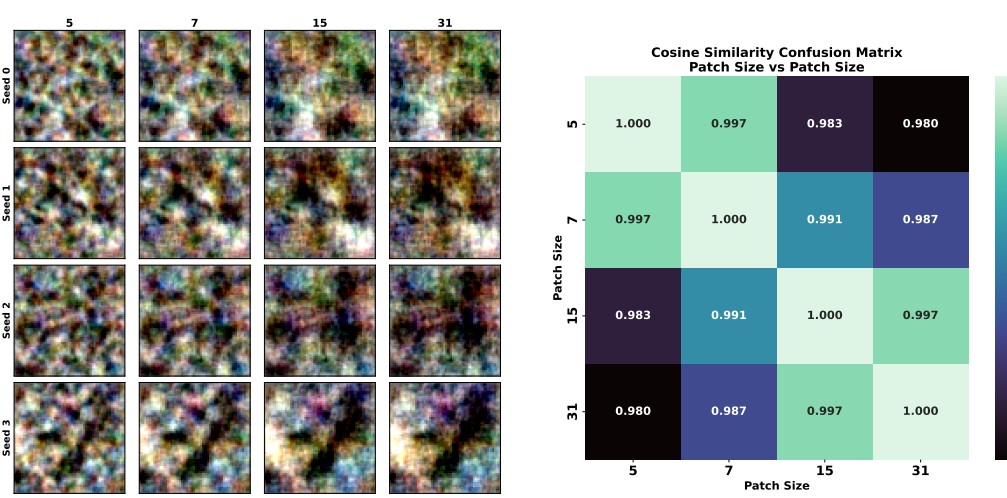

(a) Generated images from different patch sizes.

(b) Cosine similarity of the generated images.

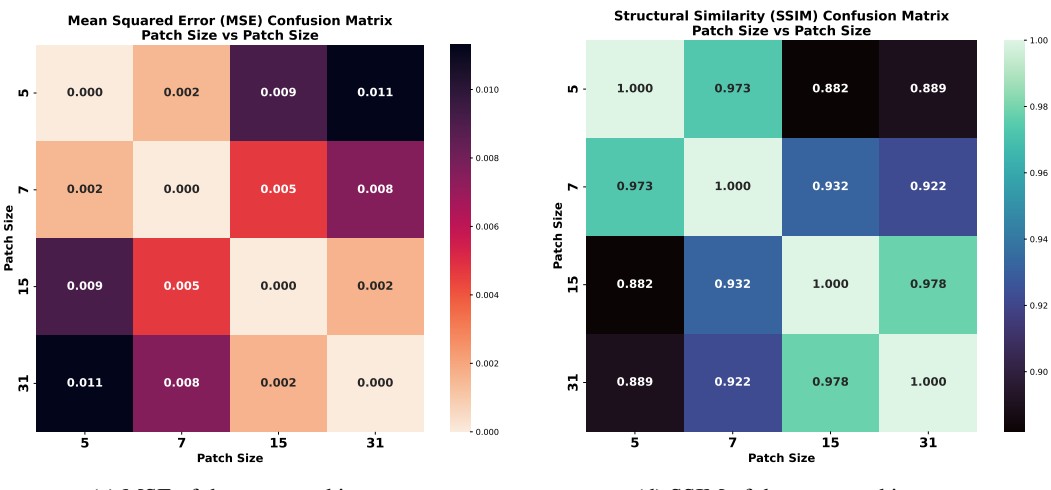

(c) MSE of the generated images.

(d) SSIM of the generated images.

Figure 12: Ablation study on the effect of patch size on constructing the PCA denoiser. (a) Generated images from different patch sizes. (b) Cosine similarity of the generated images. (c) MSE of the generated images. (d) SSIM of the generated images. The results show that the PCA denoiser is robust to the patch size used in the denoising. The generated images are similar in terms of cosine similarity, MSE, SSIM and visually. .

Table 7: Subspace alignment (mean cosine, ↑ is better) for $p$=5. First 10 directions and coarse grid.

| $k$ | 1 | 2 | 3 | 4 | 5 | 6 | 7 | 8 | 9 | 10 | Avg |
|---|---|---|---|---|---|---|---|---|---|---|---|
| Align ↑ | 0.9984 | 0.9042 | 0.9960 | 0.9969 | 0.8702 | 0.9926 | 0.9369 | 0.9410 | 0.9361 | 0.9380 | 0.9510 |

| $k$ | 10 | 20 | 30 | 40 | 50 | 60 | 70 | 80 | 90 | 100 | Avg |
|---|---|---|---|---|---|---|---|---|---|---|---|
| Align ↑ | 0.9380 | 0.8884 | 0.8739 | 0.9196 | 0.9468 | 0.9057 | 0.9409 | 0.9441 | 0.9530 | 0.9899 | 0.9300 |

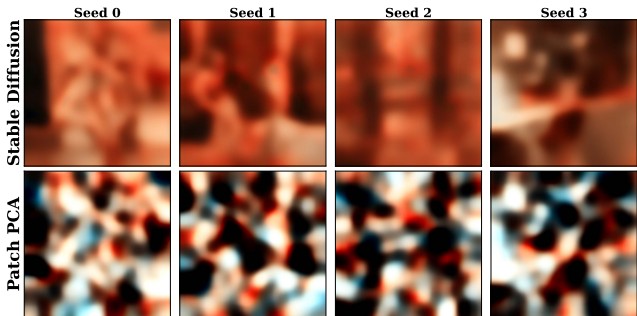

Figure 13: Stable Diffusion generated image vs. Patch PCA denoised image when projected onto the **top-2 subspace**. While structurally similar, the Stable Diffusion image projection shows a strong red-brownish color tone that is absent in the Patch PCA denoised image projection. This discrepancy likely stems from the second leading eigenvector (shown in Figure 11) having a light blue color tone that contrasts with the red-brownish tone.

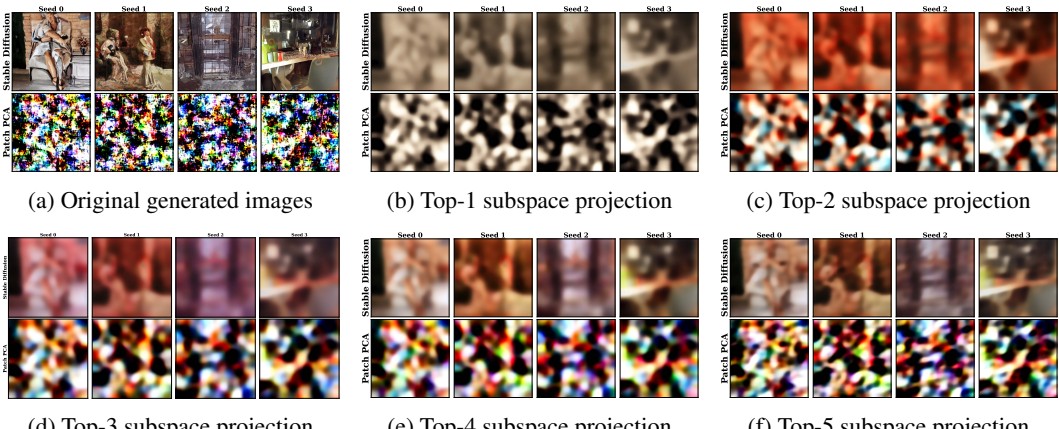

(a) Original generated images     (b) Top-1 subspace projection     (c) Top-2 subspace projection

(d) Top-3 subspace projection     (e) Top-4 subspace projection     (f) Top-5 subspace projection

Figure 14: Complete visualization of generated images from Stable Diffusion (top row in each subfigure) and Patch PCA model (bottom row) across different subspace projections. Each column shows the same noise seed (0-3). As more principal components are included in the projections (from top-1 to top-5), more detailed features are captured while maintaining the fundamental compositional structure identified in lower-dimensional projections.

mask. An example of the mask is shown in Figure 15a where the red area is the area to be edited and the blue area is the area to be preserved. We apply the same editing procedure as in the experiment in Figure 7 but only to the area defined by the mask. The results are shown in Figure 16. We observe that the localized editing resulted in a more localized composition constraint. Also, similar to the previous experiment, when we do not include the eigenvectors corresponding to the color variations in our editing procedure, the generated image Figure 16c is more similar to the reference image Figure 16a overall and only the top-right corner composition showing curved shape which aligns with the masked part of the target reference image Figure 15.

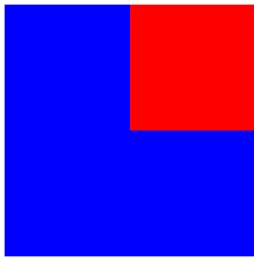

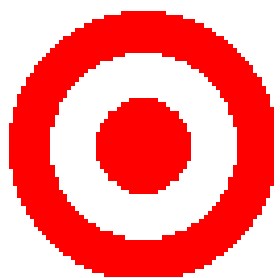

(a) Top right corner mask. The red area is edited, the blue area is preserved.

(b) Target reference image with central and surrounding compositional structure.

Figure 15: Reference images used in our editing procedure. Left: The mask defines which areas of the noise to modify. Right: The target reference image whose compositional structure we aim to transfer.

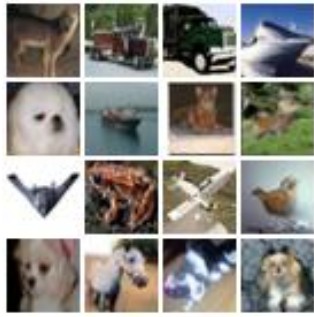

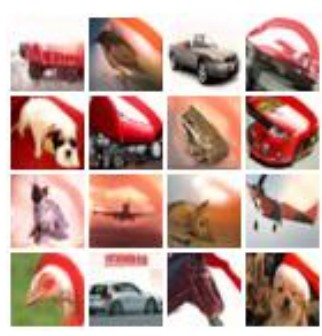

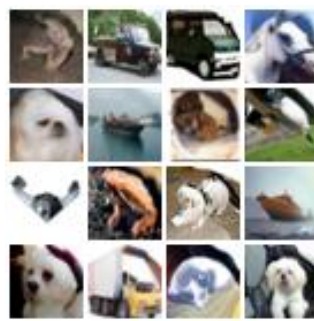

(a) Reference generation without editing.

(b) Editing in range $[1, 9]$ with mask applied.

(c) Editing in range $[1, 9] \setminus \{2, 5\}$ with mask.

Figure 16: Editing results on CIFAR-10 with target reference using interpolation factor $\alpha = 0.25$ as in the main experiments. The mask is applied to restrict editing to only the top-right corner of the image. The resulting editing is shown in Figure 16b and Figure 16c. The reference generation is shown in Figure 16a. The edited images closely align with the reference image/compositional structure in the top-right corner.

**More examples of reference images** To demonstrate our editing approach with a more complex compositional structure, we utilize Vincent van Gogh's iconic "Starry Night" painting as a reference image (shown in Figure 17, resized to $32 \times 32$). This reference image contains a strong diagonal movement from bottom left to middle right. We still use the same patch size of $4 \times 4$ as in the experiment in the main text and we choose the range $[1, 9]/\{1, 2, 5\}$ to edit the noise with interpolation factor $\alpha$ ranging from 0.0 to 1.0 ($\alpha = 0.0$ is the original (non-edited) generation). The generated images are shown in Figure 18 with initial noise seeds from 0 to 9. The generated images show a smooth transition from the original image to those with diagonal movement, like the reference Starry Night image. This shows that our editing procedure can be used to transfer the compositional structure of a complex reference image to the generated images.

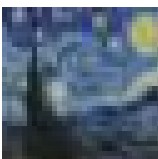

Figure 17: Reference image used for interpolation in Figure 18.

**Examples of editing in latent diffusion model** We also apply our noise editing procedure to latent diffusion models. We use patch size $5 \times 5$ with the eigenvalue spectrum shown in Figure 11a and eigenvectors visualized in Figure 11b. We use an empty prompt and the simple target-shaped image (right-hand side of Figure 6) as the reference image.

First, we compare the effects of including versus excluding color-focused eigenvectors in our editing procedure. To demonstrate the versatility and control offered by our approach, we explore a full range of interpolation factors $\alpha$ from 0.0 to 1.0 with 0.1 increments. Figure 19 shows this comparison with two frequency ranges: $[1, 12]$, which includes the top 4 color-focused eigenvectors (left), and $[5, 12]$,

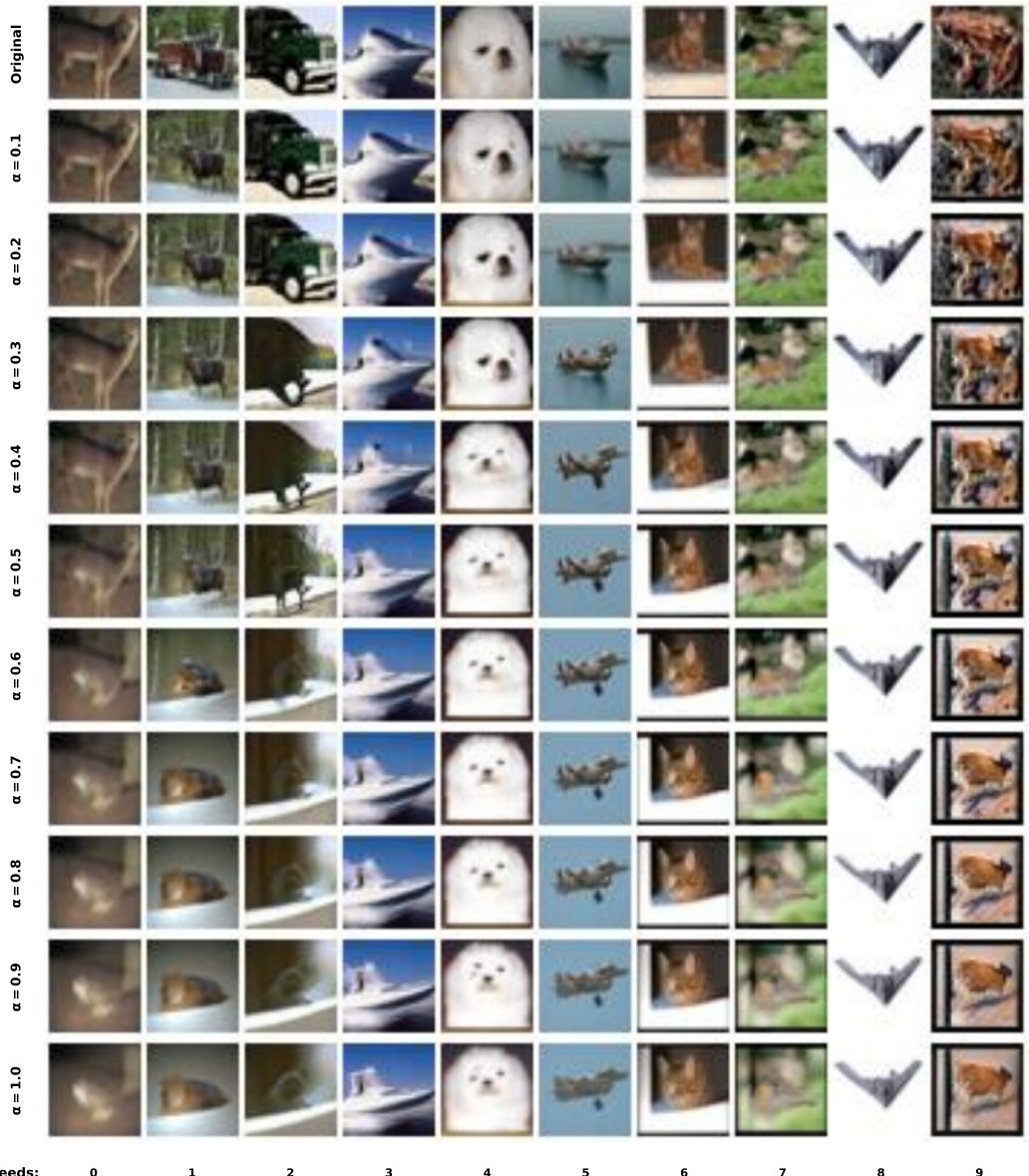

Figure 18: Generated images using the painting "Starry Night" by Vincent van Gogh as the reference image. The interpolation factor $\alpha$ ranges from 0.0 to 1.0 ($\alpha = 0.0$ is the original (non-edited) generation). The generated images show a smooth transition from the original image toward variants with diagonal movement, similar to the reference Starry Night image.

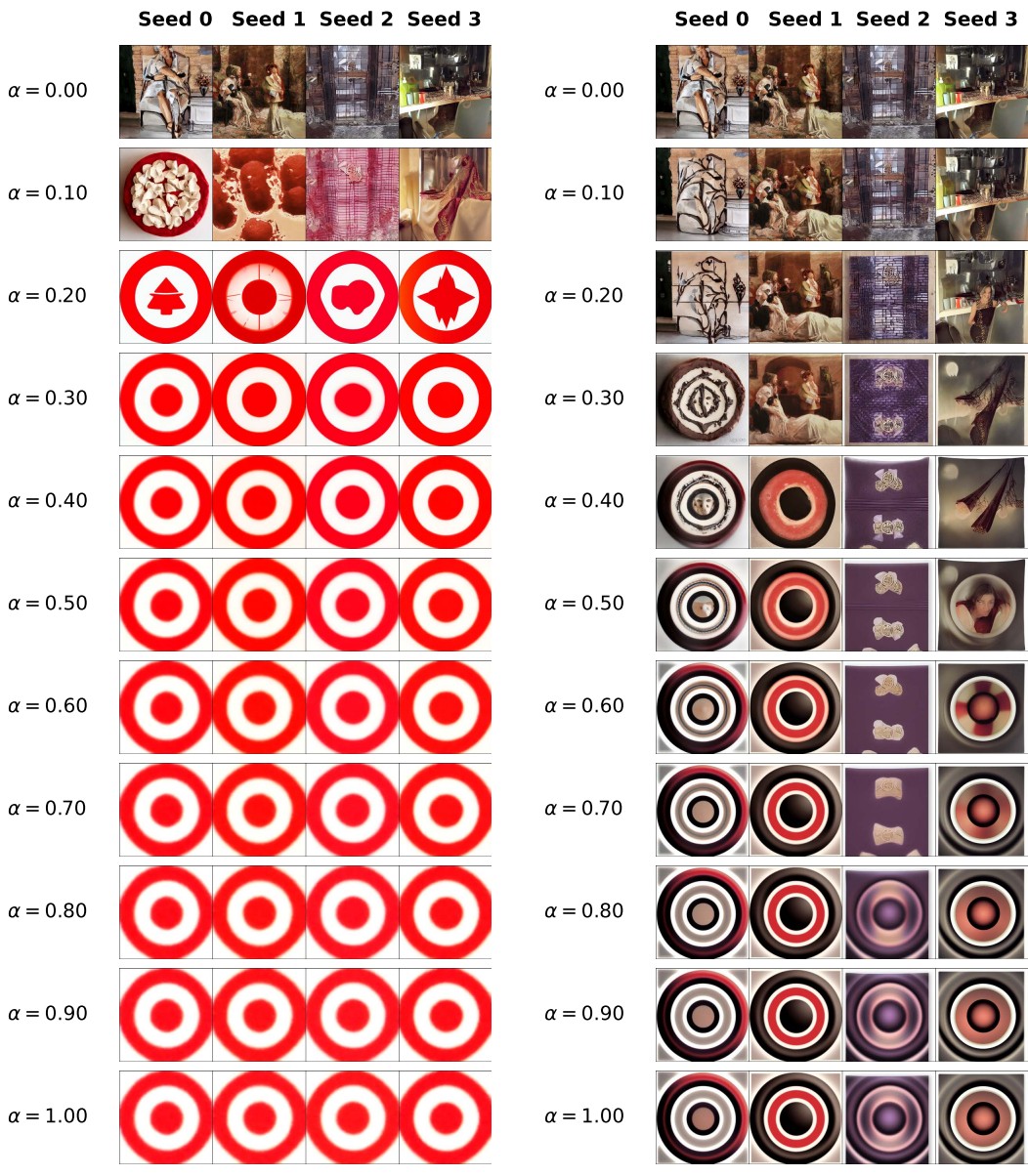

(a) Editing with range [5, 12] (excluding color-focused eigenvectors).

(b) Editing with range [1, 12] (including color-focused eigenvectors).

Figure 19: Zero-shot noise editing in latent diffusion models comparing two different frequency ranges with interpolation factor $\alpha$ ranging from $0.0$ to $1.0$. Left: Using range [1, 12] which includes color-focused eigenvectors, resulting in both structural and color transfer. Right: Using restricted range [5, 12] which excludes color-focused eigenvectors, preserving original colors while adopting the target's compositional structure.

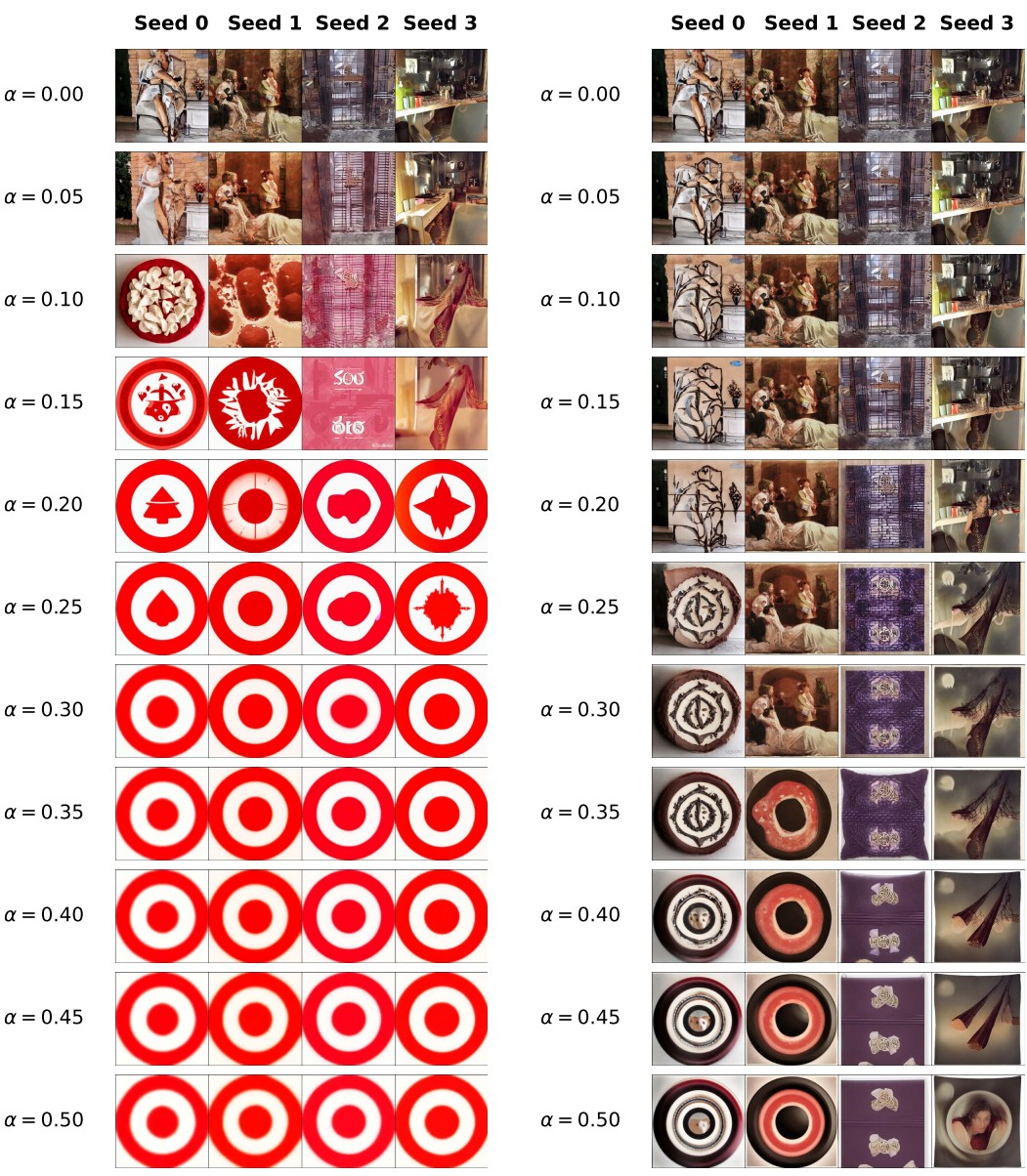

(a) Editing with range [5, 12] (excluding color-focused eigenvectors).

(b) Editing with range [1, 12] (including color-focused eigenvectors).

Figure 20: Fine-grained progression of zero-shot noise editing in latent diffusion model with interpolation factors from $\alpha = 0.0$ to $\alpha = 0.5$ with $0.05$ increments. Left: Using range [1, 12] which includes color-focused eigenvectors, resulting in both structural and color transfer. Right: Using range [5, 12] which excludes color-focused eigenvectors, preserving original colors while still adopting centered and circular compositional structure.

which excludes them (right). When color-focused eigenvectors are included, while the generated images align with the composition of the target-shaped reference image, they also exhibit a strong resemblance to the color of the reference image; see Figure 19b. When these eigenvectors are excluded, the generated images maintain their original colors while still adopting the circular and centered compositional structure of the target; see Figure 19a. This demonstrates that our approach of choosing the frequency range is effective in controlling the compositional structure and color transfer in the generated images.

To further investigate the gradual evolution of compositional structure, Figure 20 shows a more fine-grained progression of editing strength. With increasing $\alpha$ values, we observe the composition of the generated images gradually shifting toward increasingly pronounced central and circular structures, even though the overall image does not change much, especially when the color-focused eigenvectors are excluded (Figure 20a).

**Noise editing in high-frequency bands.** While our main experiments focus on low-frequency components for composition control, our Patch-PCA based noise editing method naturally extends to high-frequency bands. We test this by using *Starry Night* as the reference image (Figure 17) and a $5 \times 5$ Patch-PCA basis. The reference image contains blue-yellow oil-painting texture and curly stylistic texture. We observe that by editing the high-frequency coefficients, we can achieve a texture transfer effect. This effect can be sensitive to the choice of frequency band to determine what type of texture to transfer. A visual example is shown in Figure 21. When we use the frequency band [19, 75] and with a small interpolation factor of 0.3, we already see the blue-yellow oil-painting texture in the generated images. When we use the frequency band [20, 75], the blue-yellow oil-painting texture is not as easy to generate, and we mostly pick up the curly stylistic texture even with a large interpolation factor of 0.9. In all cases, high-frequency band noise editing leaves object layout largely unchanged while altering texture, complementing the low-frequency compositional control. A more comprehensive study across the frequency bands is left for future work.

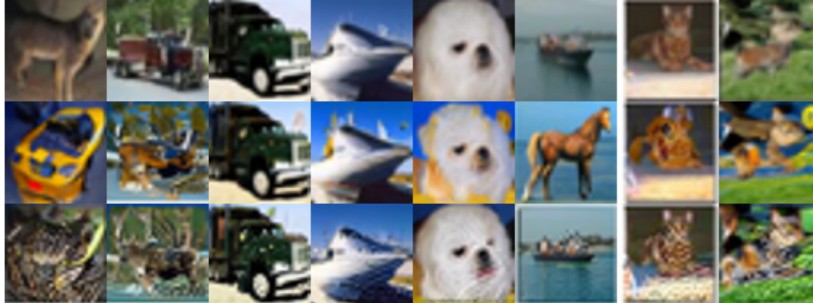

Figure 21: Noise editing in high-frequency bands results using frequency band [19, 75] and [20, 75] with interpolation factor $\alpha = 0.3$ and $\alpha = 0.9$ respectively.

**Editing for Object Count (Three-Object Composition).** To test whether our noise editing method can affect object *count*, we use the ImageNet class "granny smith apple" (class 948) and create a reference image containing three circular white blobs on a black background. We design the task for achieving three apples in the generated images. We edit the initial noise in the same way as the ImageNet editing experiment in Figure 8, i.e., using low-frequency Patch-PCA bands $[1, 9] \setminus \{1, 2, 5\}$.

*Results.* Before editing, most samples contain either only one apple or more than three apples. After editing, $43/64$ samples exhibit the intended three apples near the highlighted areas in our reference image, confirming that low-frequency structure can influence object counting (see Figure 23). However, we also observe that the background diversity is almost removed. We leave it as future work to balance the object count and the background diversity.

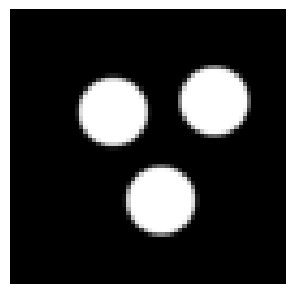

Figure 22: Three-object reference used for count alignment.

Table 8: Distribution of object counts before and after low-frequency editing (64 seeds).

| # Apples | 0 | 1 | 2 | 3 | 4+ |
|---|---|---|---|---|---|
| No Edit | 5 | 30 | 8 | 5 | 16 |
| Edited | 2 | 8 | 11 | 43 | 0 |

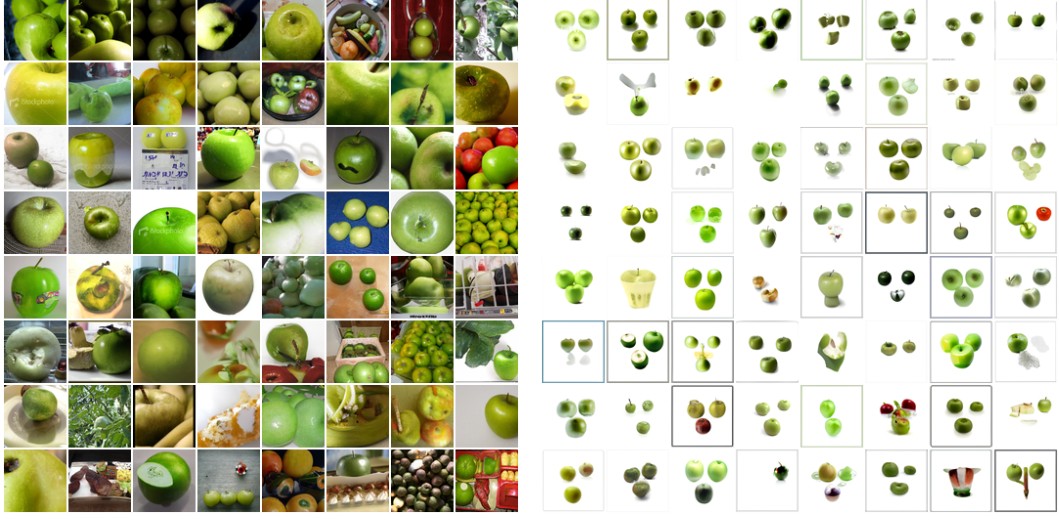

(a) No Edit (64 seeds).  (b) Edited (bands $[1, 9] \setminus \{1, 2, 5\}$, $\alpha=0.22$).

Figure 23: Object-count control via low-frequency editing. Left: unedited generations, most with one apple or more than three apples. Right: edited generations, most with three apples in the area of the of circular white blobs of the reference image.

## A.5 Datasets and pre-trained models

The datasets used in our experiments are ImageNet [8], FFHQ [16], AFHQ [6], and CIFAR-10 [18]. The ImageNet dataset contains 1.28 million images with 1,000 classes. The FFHQ dataset contains 70,000 high-quality images of human faces. The AFHQ dataset contains 15,000 high-quality images of animals (cats, dogs, and wild animals). The CIFAR-10 dataset contains 60,000 images in 10 classes, with 6,000 images per class.

The pre-trained models used in our experiments are the diffusion models trained on ImageNet, FFHQ, and AFHQ datasets are available from the official repository of Karras et al. [17]: `https://github.com/NVlabs/edm` and the transformer-based U-ViT model is available from the official repository of Bao et al. [4]: `https://github.com/baofff/U-ViT`. We also train flow matching on the AFHQ dataset using the repository of Tong et al. [31]: `https://github.com/atong01/conditional-flow-matching`.

