# OpenReview forum: "Seeds of Structure: Patch PCA Reveals Universal Compositional Cues in Diffusion Models"
_NeurIPS.cc/2025/Conference — NeurIPS 2025 poster_

### Official Review · Reviewer_1av2 · 2025-06-26

**Clarity:** 3
**Significance:** 1
**Originality:** 2
**Rating:** 3
**Confidence:** 4

**Summary:**

The paper presents a methodology to understand the relationship between the initial noise and generated images in diffusion models. Using the principal components of the training dataset patches, the authors find that low-frequency structures in the initial noise are also retained in the generated images. The authors aim to exploit this finding and propose a zero-shot editing algorithm that attempts to modify these structures in the initial noise and control the composition of the final generated image.

**Questions:**

- Does the proposed Patch PCA translate to diffusion transformers (DiTs)? The convolutional models utilize patch-specific features to perform denoising, but diffusion transformers are not biased towards using local patches. Including DiTs in the analysis could help drive the main argument of the paper further.
- Why did you choose ImageNet for extracting the principal components, and how would using a different dataset (e.g. FFHQ) affect the analysis and editing?  Since FFHQ is mostly aligned, with faces centered on the images, could you extract different dataset-specific components that the diffusion models learned to compose (e.g. eyes, mouth)?

**Ethical Concerns:**

["NO or VERY MINOR ethics concerns only"]

**Final Justification:**

The paper offers an interesting analysis of how the initial noise in diffusion models affects the contents of the final generated image using the PCA of image patches. The authors also propose an editing algorithm that utilizes the low-frequency structures in the initial noise to guide image generation with a reference image. The image editing algorithm has not been evaluated extensively and would benefit from further experiments on additional models/datasets.

**Limitations:**

The only limitation mentioned is that the proposed editing is focused on low-frequency components. However, the most significant limitations would be regarding the applicability of the proposed editing, which has not been discussed thoroughly. The authors should expand their experimental section to demonstrate and address potential limitations of the proposed editing algorithm.

**Paper Formatting Concerns:**

None.

**Quality:**

2

**Strengths And Weaknesses:**

**Strengths**:
- The main finding of the paper is that the commonly used convolutional diffusion models utilize specific structures in the initial noise when generating images. The authors utilize the principal components of the training dataset patches to probe the initial noise for these patterns and showcase how the diffusion-generated images exhibit similar patterns to the ones highlighted by the PCA components applied on the initial noise. The analysis is straightforward and provides an elegant explanation to an empirical observation that has been previously made regarding the properties of the initial noise and how they are correlated with the generated final image.

**Weaknesses**:
- The authors aimed to exploit their findings and propose an editing scheme using the extracted PCA components. The PCA-based noise editing requires considerably more experimentation to support and, in the end, hurts the overall presentation of the paper. The authors claim that the proposed zero-shot Patch PCA editing enables compositional control in the generated images. However, the experiments to support this are limited, with the only example being the interpolation between generated images and a 'Target' reference image. The reference image has a very distinct structure, which, although it seems to affect the CIFAR-10 and ImageNet generations, in no way guarantees that the proposed algorithm works in a more general setting. In the supplementary, the authors demonstrate editing with another reference image, but the results in Fig. 18 are nowhere near as successful. To claim compositional control, a larger-scale experiment is necessary, such as the one performed by Li et al. [11], which does numerical composition, i.e., generating K distinct objects in the image.

---

> ### Author Rebuttal · Authors · 2025-07-31
>
> We thank the reviewer for the feedback and constructive questions, and for recognizing our analysis that explains how the initial noise correlates with the generated final image.
>
> ### **1. On the scale of the editing experiments.**
> The only weakness raised is the scale of the editing experiments.
> >The authors aimed to exploit their findings and propose an editing scheme using the extracted PCA components. The PCA-based noise editing requires considerably more experimentation to support and, in the end, hurts the overall presentation of the paper. The authors claim that the proposed zero-shot Patch PCA editing enables compositional control in the generated images. However, the experiments to support this are limited, with the only example being the interpolation between generated images and a 'Target' reference image. The reference image has a very distinct structure, which, although it seems to affect the CIFAR-10 and ImageNet generations, in no way guarantees that the proposed algorithm works in a more general setting. In the supplementary, the authors demonstrate editing with another reference image, but the results in Fig. 18 are nowhere near as successful. To claim compositional control, a larger-scale experiment is necessary, such as the one performed by Li et al. [11], which does numerical composition, i.e., generating K distinct objects in the image.
>
> We think there might be a misunderstanding of our paper's core contribution compared to works like Li et al. [11], which we hope to clarify.
>
> Our paper's primary contribution is a fundamental analysis of the image diffusion model: we identify that low-frequency components of initial noise are key to compositional control. As the reviewer noted, this provides an "elegant explanation" of the noise-to-image correlation.
>
> Li et al. [11] leverage the intuition of reliable noise and focus on the downstream application of mining reliable seeds to improve text-to-image generation. While large-scale experiments are crucial for their goal, our editing method is intended as proof-of-concept—a clear demonstration of our core finding about Patch PCA components of initial noise. Its success validates our main thesis about frequency bands.
>
> We clarify that we define image composition in Lines 32-33 as illumination and major structural lines that contribute to perceived composition, which can be measured by SSIM. The numerical composition is a completely different concept. Whether some principal components from Patch PCA reflect numerical composition is an interesting topic for future study.
>
> In our response to other reviewers, we have incorporated feedback to:
> 1. Add SSIM scores and color histogram correlation to quantitatively evaluate compositional alignment and color preservation (Item 1 in rebuttal to Reviewer t5kH)
> 2. Include new experiments on high-frequency noise editing influence (Item 2 in rebuttal to Reviewer t5kH)
> 3. Verify findings hold across different deterministic samplers like VE and VP (Item 3 in rebuttal to Reviewer HRUH)
> 4. Verify findings hold for flow-matching models (Item 4 in rebuttal to Reviewer HRUH)
>
> Given that our paper's central scientific contribution is distinct from [11] and the strengthened evaluation, we believe our work makes a valuable contribution to understanding diffusion models.
>
> **Clarification on Compositional Control Success in Figure 18**
> >In the supplementary, the authors demonstrate editing with another reference image, but the results in Fig. 18 are nowhere near as successful
>
> We respectfully disagree that editing in Fig. 18 is unsuccessful. For Starry Night, the main perceived structure is diagonal movement from bottom left to middle right. Generated images with increasing weights for the reference show a transition from original generation toward variants with diagonal movement similar to the reference, showing compositional structure alignment. We conducted SSIM computation comparing baseline (no edit) and edited case with interpolation factors, showing a monotonic increasing trend:
>
> | Interpolation factors | 0.0 (no edit) | 0.1 | 0.2 | 0.3 | 0.4 | 0.5 | 0.6 | 0.7 | 0.8 | 0.9 | 1.0 |
> |---------------------|---------------|-----|-----|-----|-----|-----|-----|-----|-----|-----|-----|
> | SSIM vs. Reference  | 0.0272        | 0.0741 | 0.1217 | 0.1707 | 0.1915 | 0.2145 | 0.2208 | 0.2309 | 0.2386 | 0.2526 | 0.2614 |
>
> This quantitatively demonstrates successful progressive alignment toward reference composition.
>
> ### **2. On Translating Patch PCA to DiTs**
> >Does the proposed Patch PCA translate to diffusion transformers (DiTs)? The convolutional models utilize patch-specific features to perform denoising, but diffusion transformers are not biased towards using local patches. Including DiTs in the analysis could help drive the main argument of the paper further.
>
> Thank you for this insightful question. Yes, our findings and methods apply to transformer-based diffusion architectures like DiTs. Transformer-based diffusion models, such as DiT, have an initial "Patchify" step where the image is broken into tokens, creating an inductive bias compatible with our patch-based analysis. Additionally, as shown in recent work [24], different architectures—including U-Nets, U-ViT, and Consistency Trajectory Models—produce remarkably nearly identical images when given the same initial noise (see Fig. 1 and Fig. 11 in [24]). Consequently, both frequency perturbation phenomenon and editing apply to diffusion architecture with transformers that we empirically verify below.
>
> We couldn't find the DiT pre-trained model on ImageNet-64, FFHQ, or AFHQ, so we used U-ViT instead. We generated 1000 images with identical seeds for EDM and U-ViT, both of which are publicly available pre-trained models on ImageNet and Patch PCA, as well as edited versions using the same edited initial noises, following Figure 1 settings with an interpolation factor of 0.1. We show SSIM and MSE scores between the same seed (paired) and random (unpaired) images.
>
> | Network Pair | SSIM Paired | SSIM Random | MSE Paired | MSE Random |
> |--------------|-------------|-------------|------------|------------|
> | U-ViT vs EDM | 0.8197 ± 0.0794 | 0.0811 ± 0.0522 | 0.0065 ± 0.0045 | 0.1422 ± 0.0657 |
> | U-ViT vs Patch PCA | 0.4805 ± 0.1265 | 0.0266 ± 0.0248 | 0.0542 ± 0.0263 | 0.1248 ± 0.0371 |
> | U-ViT Edited vs EDM Edited | 0.8093 ± 0.0838 | 0.1219 ± 0.0645 | 0.0102 ± 0.0094 | 0.1703 ± 0.0732 |
>
> These results confirm our Patch PCA based observations and that our editing method applies across different architectures, with edited images showing strong structural alignment regardless of underlying model architecture.
>
> ### **3. On the Dataset for Patch PCA Computation**
>
> The specific dataset used to compute Patch PCA basis is not critical for small patch sizes we use. In Appendix Figure 9, we show that patch covariance matrices computed from ImageNet, FFHQ, and AFHQ are nearly identical for patches smaller than 16×16 (achieving >0.99 correlation). This is because small patches capture universal, low-level statistics of natural images that transcend specific content of any dataset. When using small patches, high-level structures like aligned faces in FFHQ are broken down into generic components (edge detectors, textures), making their statistics highly similar to those from general-purpose datasets like ImageNet. Therefore, we did not observe components specific to features like mouths or eyes.
>
> ### **4. On Limitations**
> >The only limitation mentioned is that the proposed editing is focused on low-frequency components. However, the most significant limitations would be regarding the applicability of the proposed editing, which has not been discussed thoroughly. The authors should expand their experimental section to demonstrate and address potential limitations of the proposed editing algorithm.
>
> As we clarified earlier, our applicability to Fig. 18 is confirmed by SSIM scores. This paper is intended as a proof-of-concept—a clear demonstration of our core finding about Patch PCA components of initial noise. A comprehensive evaluation remains a future goal.
>
> ---
>
> [11] Li, Shuangqi, Hieu Le, Jingyi Xu, and Mathieu Salzmann. "Enhancing compositional text-to-image generation with reliable random seeds." ICLR 2024.
>
> [24] Huijie Zhang, Jinfan Zhou, Yifu Lu, Minzhe Guo, Peng Wang, Liyue Shen, and Qing Qu. The emergence of reproducibility and consistency in diffusion models. ICML 2024.

---

> > ### Comment · Reviewer_1av2 · 2025-08-03
> >
> > Thank you for your detailed responses.
> >
> > > On the scale of the editing experiments.
> >
> > Li et al. [11] seem to suggest that if the original noise contains some low-frequency structure that separates the K individual objects well, then the generated image will also have K well-separated objects. I believe that a similar experiment is necessary to show that your proposed PCA analysis could be valuable for editing tasks. The proposed editing algorithm is limited, and no relevant baselines have been provided even after the rebuttal, making it difficult to contextualize the results given.
> >
> > > Clarification on Compositional Control Success in Figure 18
> >
> > I am assuming that you refer to the edge in the middle of the image. In that case, I would partially agree that some of the samples shown in Figure 18 follow the same edge. However, again, this editing scheme offers very limited control, and I am struggling to see what an actual application would be where a user would want to perform these kinds of edits. This kind of mixing between low and high-frequency components of images with diffusion models has been shown before (ILVR: Conditioning Method for Denoising Diffusion Probabilistic Models, ICCV 2021).
> >
> > Overall, the image editing algorithm proposed in the paper comes across as more of an afterthought of the patch PCA analysis. More experiments and evaluations are necessary to make a compelling case for it.
> >
> > Given that my other concerns have been addressed, I will be increasing my score.

---

> > > ### Author Response · Authors · 2025-08-04
> > >
> > > We sincerely thank the reviewer for engaging deeply with our work and for increasing their score. We're glad that our earlier response addressed many of your concerns and appreciate the opportunity to clarify the role of our editing method further.
> > >
> > > **On the Role of the Editing Method**
> > >
> > > We want to reiterate that our editing algorithm serves as a proof-of-concept to demonstrate our core analytical finding: that low-frequency components in the initial noise encode universal compositional structure, as revealed by Patch PCA. The successful compositional alignment (illumination, major edges) suggests that diffusion models extend beyond reproducing training examples to combining structural constraints with learned semantic features. The editing method itself is novel, practical, training-free, and shows potential for further exploration.
> > >
> > > **Object Count via Low-Frequency Editing**
> > >
> > > We appreciate your suggestion on extending compositional alignment from major edges to object count, related to Li et al. [11]. The visual exploration mentioned in [11] involves object-token associations via cross-attention, which our framework does not directly address. In our case, the most natural approach is to use a reference image with highlighted areas and boundaries that match the target object shape.
> > >
> > > We performed an experiment on the ImageNet dataset conditioned on class 948 (*Granny Smith apple*). Specifically, we constructed a **reference image** consisting of **three well-separated circular white blobs** on a black background and used it as the source for low-frequency editing (via interpolation with a factor of 0.22 and following the same setting in Figure 8 in the submission).
> > >
> > > Among 64 generated samples (seeds 0-63), the object count distribution shifted dramatically toward the target of three apples.
> > >
> > > **Before editing:**
> > > | **Count** | 0 | 1 | 2 | 3 | 4+ |
> > > |-----------|---|---|---|---|-----|
> > > | **Frequency** | 5 | 30 | 8 | 5 | 16 |
> > >
> > > **After editing:**
> > > | **Count** | 0 | 1 | 2 | 3 |
> > > |-----------|---|---|---|---|
> > > | **Frequency** | 2 | 8 | 11 | 43 |
> > >
> > > This demonstrates successful steering toward three-object compositions, with 43/64 samples (67%) achieving the target count. We observed that different seeds have varying sensitivity to interpolation factors—while 0.22 works well overall, larger factors often produce unrecognizable results. We will include these results and visualizations in the revised version.
> > >
> > > **Distinction from ILVR**
> > >
> > > Thank you for pointing out ILVR [Choi21]. ILVR empirically demonstrates that **iterative** low-frequency conditioning during DDPM sampling (i.e., with a **stochastic sampler**) can steer image generation. In contrast, our work focuses on **deterministic ODE-based samplers** and provides a mechanistic explanation for why low-frequency structure matters: we show that compositional cues are already embedded in the **initial noise**, and can be revealed and manipulated via Patch PCA, supported by Gaussian patch assumptions. This offers a new perspective: the low-frequency part in noise encodes structural information of the final generation, explaining previously observed correlations between seeds and generated image.
> > >
> > > Thank you again for your constructive engagement.
> > >
> > > [Choi21] Choi J, Kim S, Jeong Y, Gwon Y, Yoon S. ILVR: Conditioning method for denoising diffusion probabilistic models. ICCV 2021.

---

### Official Review · Reviewer_azVL · 2025-07-01

**Clarity:** 3
**Significance:** 3
**Originality:** 3
**Rating:** 4
**Confidence:** 3

**Summary:**

This paper investigates how initial noise structure influences the final output of diffusion models. Using patch-wise Principal Component Analysis (PCA), the authors empirically show that low-frequency components of the noise seed predominantly determine the compositional structure of the generated image. Based on this insight, the paper demonstrates that these compositional cues are universal across different models and datasets and proposes a simple Patch PCA denoiser and a zero-shot editing method to control image composition.

**Questions:**

I would like to see more theoretical analysis, and more exploration on high-frequency noise.

**Ethical Concerns:**

["NO or VERY MINOR ethics concerns only"]

**Final Justification:**

Thanks to the author for the response. Some of my questions have been resolved. Therefore, I will maintain my original score.

**Limitations:**

yes

**Quality:**

3

**Strengths And Weaknesses:**

Strengths:
1. The paper provides a strong and intuitive investigation into the noise-to-image map of diffusion models, offering a clear link between low-frequency noise and image composition. This provides valuable insight into the underlying mechanics of the generative process.
2. A key strength is the compelling empirical evidence that identical noise seeds produce structurally similar images across different models and datasets. This finding of "universal compositional cues" is significant and well-supported by both quantitative (SSIM/MSE) and qualitative results.
3. The proposed zero-shot editing method is a practical and valuable application of the paper's findings. It offers an efficient way to guide image composition without needing any model retraining or fine-tuning, which is a significant advantage over many existing editing techniques.

Weaknesses:
1. The analysis focuses heavily on the role of low-frequency components for composition, while the influence of high-frequency noise components on texture and fine details is mentioned but not explored in depth. A more complete picture would involve a deeper analysis of how different frequency bands contribute to various aspects of the final image.
2. The theoretical justification relies on a simplified Patch PCA denoiser, which, while effective, may not fully capture the complex, non-linear dynamics of large-scale neural network denoisers. This simplification could limit the direct applicability of the theoretical claims to state-of-the-art diffusion models.
3. While the editing results on CIFAR-10 and ImageNet are promising, the method's effectiveness on more complex scenes with multiple objects or intricate layouts is not fully demonstrated. The compositional control appears to be primarily global, and its precision in more complex scenarios remains an open question.

---

> ### Author Rebuttal · Authors · 2025-07-31
>
> We thank the reviewer for the encouraging assessment and for highlighting both the empirical insight of universal compositional cues and the practical value of our zero-shot editing method. Below, we respond to each weakness and question.
>
> ### **1. The Influence of High-Frequency Noise**
> >The analysis focuses heavily on the role of low-frequency components for composition, while the influence of high-frequency noise components on texture and fine details is mentioned but not explored in depth. A more complete picture would involve a deeper analysis of how different frequency bands contribute to various aspects of the final image.
>
> We agree that a comprehensive exploration of all frequency bands would be valuable. However, in this paper, our focus is on establishing the dominant structural role of low frequencies. To demonstrate that the editing method naturally extends to the high-frequency region, we conducted preliminary experiments on high-frequency perturbation. Using "Starry Night" as the reference image (Figure 17, appendix) with 5×5 Patch PCA basis, we find:
>
> 1. **Frequency band [19:75], interpolation factor 0.3**: Creates blue and yellow oil-painting texture like in Starry Night while maintaining the original composition.
> 2. **Frequency band [20:75], interpolation factor 0.9**: Generated image remains almost identical to the unedited version except for additional curly textures mimicking Starry Night's swirling patterns.
>
> These experiments demonstrate that our method can effectively manipulate texture. We will include these results in the updated paper.
>
> ### **2. Theoretical Justification for Neural Network Denoisers**
> >The theoretical justification relies on a simplified Patch PCA denoiser, which, while effective, may not fully capture the complex, non-linear dynamics of large-scale neural network denoisers. This simplification could limit the direct applicability of the theoretical claims to state-of-the-art diffusion models.
>
> We appreciate the opportunity to clarify the relationship between our theoretical analysis and practical neural network denoisers.
>
> Our theoretical framework handles nonlinear settings. Our main result, Theorem A.2, provides closed-form expressions for the optimal denoiser within a broad family of "admissible local functions," which includes both linear and nonlinear functions. This family is motivated by the local score and equivariant score functions introduced in [2] and offers a natural generalization to settings where equivariance is broken—for example, near boundaries. Notably, our analysis reveals that the equivariance condition is redundant; the key structural constraint is what we define as the "patch shape" in Definition A.1.
>
> Theorem 5.1 follows as a corollary of Theorem A.2, establishing that when the patch distribution is Gaussian, the optimal denoiser within the admissible local function family coincides with the Patch PCA denoiser. This suggests that even complex neural networks, if locally determined, should approximate this theoretical optimum for Gaussian-like patch distributions during training.
>
> For neural network denoisers trained on non-Gaussian but approximately Gaussian patch distributions (e.g., close in Wasserstein distance), we hypothesize that the neural network denoisers can be viewed as a bounded perturbation of the optimal Patch PCA denoiser. Based on this assumption, if we denote the neural network denoiser as $f_{NN}$ and Patch PCA denoiser as $f_{PCA}$, assuming $||f_{NN}(x, t) - f_{PCA}(x, t)||_2 \leq \epsilon(t)$ and bounds on the Lipschitz constant $L_t$ of flow maps $\Psi_t$, we can adapt the Alekseev–Gröbner stability argument from [1] to bound trajectory discrepancies. This framework implies that final samples from neural denoisers remain close to those from our theoretical model when the assumptions are satisfied.
>
> ### **3. Editing on More Complex Scenes**
> >While the editing results on CIFAR-10 and ImageNet are promising, the method's effectiveness on more complex scenes with multiple objects or intricate layouts is not fully demonstrated. The compositional control appears to be primarily global, and its precision in more complex scenarios remains an open question.
>
> We agree that extending the method to multi-object scenes and finer spatial precision is compelling future work. Our primary goal here is to introduce a new Patch PCA filter-based perspective for editing via single manipulation of the initial seed.
>
> For reference images more complex than the target image, Figure 18 in the Appendix demonstrates editing results for CIFAR-10 using Starry Night as the reference, where our method successfully captures the characteristic bottom-left to middle-right diagonal movement of the reference composition. We will leave a comprehensive evaluation of more complex multi-object scenes as an important direction for future studies.
>
> ---
>
> [1] Benton, Joe, George Deligiannidis, and Arnaud Doucet. "Error Bounds for Flow Matching Methods." TMLR 2024
> [2] Kamb, and Ganguli. "An analytic theory of creativity in convolutional diffusion models". ICML 2025

---

### Official Review · Reviewer_HRUH · 2025-07-03

**Clarity:** 3
**Significance:** 3
**Originality:** 4
**Rating:** 5
**Confidence:** 3

**Summary:**

The paper presents an empirical study on the latent noise variables used to seed image-diffusion model generations. The authors use patch-PCA to find that lower frequency components of the noisy latent can strongly influence the compositional structure and semantics of the image while higher frequency components have a smaller impact instead affecting attributes like texture. The authors then propose a zero-shot image editing technique using the PCA frequency components that enables style transfer.

**Questions:**

[L114] Can you please describe line 114 in more detail? i.e. “resampling the identified frequency bands [n,48] on each patch.” An algorithm like algorithm 1 would be nice (can be included in the appendix).
[Fig 1b] what do the three rows correspond to?
[L132] Are different models are being used here? Are you using an ImageNet trained model A and a FFHQ trained model B and then generating images with the same seed?
[Tab 1, Fig 5] How do the differences measure in terms of the LPIPS metric?
Was a patch size of 4x4 used for all experiments? How do results from the noise editing algorithm differ when using different patch sizes?

**Ethical Concerns:**

["NO or VERY MINOR ethics concerns only"]

**Final Justification:**

I maintain my recommendation for acceptance.

This is an interesting research paper with non-trivial findings. My reasons are outlined in the strengths section, and most of my concerns have been addressed in the rebuttal. I think this work's findings should be shared with the NeurIPS community.

**Limitations:**

The authors do discuss limitations at the end of the main text, which I think fairly summarizes the shortcomings of this study.

**Paper Formatting Concerns:**

None.

**Quality:**

4

**Strengths And Weaknesses:**

Strengths:
**(S1)**: The paper does a thorough and interesting investigation into the compositional patterns of noise for diffusion models. The findings are non-trivial and relevant to the broader field as they show that diffusion models learn compositional features and data structure across different “frequencies” and that this is highly correlated with the initial noisy latent across datasets and models.
**(S2)**: The study generalizes to latent-diffusion models which are more widely used.
**(S3)**: The paper proposes a new zero-shot editing technique that can be used for image or style translation tasks based on the empirical findings with their patch-pca investigation.

Weaknesses:
**(W1)**: Zero-shot patch pca based noise editing is missing quantitative comparisons. I’m very curious about how this compares to methods like SDEdit, and if some of the insights of this technique can apply to SDEdit or vice-versa. It would be nice to show comparison with other zero-shot image and style editing techniques as well.
**(W2)**: How does the choice of sampler affect the findings of the paper? If I’m not mistaken, the study is currently using DDIM. Does this change for DDPM (given the stochasticity of each denoising step) or other VE/VP samplers?
**(W3)**: How would this analysis change for flow-matching based models? I don’t see a discussion of this, but I’m curious to know if similar patterns emerge.

Overall, this paper is very interesting and presents findings that should be relevant to the broader diffusion community. It is clearly written and the investigation seems sound. I would recommend it for acceptance.

---

> ### Author Rebuttal · Authors · 2025-07-30
>
> We thank the reviewer for the constructive feedback and are encouraged by your recognition of the non-trivial nature of our findings and the practical value of the proposed zero-shot editing method. We address the questions below.
>
> ### **1. Quantitative Evaluation of Zero-Shot Editing**
> >Zero-shot patch pca based noise editing is missing quantitative comparisons.
>
> We agree that adding quantitative evaluation would further strengthen the experiments. To complement the qualitative results in Figure 7, we added quantitative evaluations across 256 random seeds using CIFAR-10. We use SSIM for structure alignment and Color Histogram Correlation for global color similarity. Color Histogram Correlation compares normalized RGB histograms using Bhattacharyya distance (range: 0 to 1, where 1 indicates identical distributions).
>
> | Metric | (a) No Edit | (b) Edit [1,9] | (c) Edit [1,9] \ {2,5} |
> |--------|-------------|----------------|------------------------|
> | SSIM vs. Ref ↑ | 0.0844 ± 0.0708 | 0.2521 ± 0.0665 | 0.2887 ± 0.0812 |
> | Color Hist. Corr. vs. Ref ↓ | 0.0402 ± 0.0860 | 0.2515 ± 0.0827 | 0.1110 ± 0.0987 |
> | Color Hist. Corr. vs. No Edit ↑ | N/A | 0.3809 ± 0.1137 | 0.5759 ± 0.1130 |
>
> Results show that editing low-frequency Patch PCA components significantly improves SSIM, confirming better compositional alignment. Excluding color channels (c) achieves better color preservation, as demonstrated by higher correlation with the no-edit case and lower correlation with the reference image. We will add these results to the main text.
>
> ### **2. Relation to SDEdit and Other Zero-Shot Editors**
> >I'm very curious about how this compares to methods like SDEdit, and if some of the insights of this technique can apply to SDEdit or vice-versa. It would be nice to show comparison with other zero-shot image and style editing techniques as well.
>
> SDEdit interpolates a reference image with noise at selected timesteps, then denoises from that intermediate state. Our method manipulates Patch PCA frequency bands in the initial noise. These approaches introduce editing at different aspects: SDEdit at selected timesteps, ours via frequency bands. Combining our approach with SDEdit for finer-grained control is an interesting direction. Unfortunately, this year's policy does not allow us to present visual results, and we will leave it as future work. We agree that a full benchmark across major zero-shot editors would be valuable. However, it falls outside this paper's scope, as our primary goal is to expose the structural role of low-frequency noise and introduce a novel, lightweight, training-free editing tool. We will leave comprehensive benchmarking as future work.
>
> ### **3. Sampler Effect**
> >How does the choice of sampler affect the findings of the paper? If I'm not mistaken, the study is currently using DDIM. Does this change for DDPM (given the stochasticity of each denoising step) or other VE/VP samplers?
>
> Yes, our study uses DDIM (EDM scheduling). Other deterministic samplers (VE, VP) produce identical results for the same model and initial noise when given sufficient steps. Our observations and methods apply to all deterministic samplers, empirically verified by near-identical images across samplers:
>
> | Sampler Pair | SSIM | MSE |
> |--------------|------|-----|
> | EDM vs VE | 0.990235 ± 0.027491 | 0.000236 ± 0.000976 |
> | EDM vs VP | 0.982148 ± 0.038618 | 0.000590 ± 0.001448 |
> | VP vs VE | 0.989944 ± 0.015296 | 0.000224 ± 0.000444 |
>
> However, stochastic samplers like DDPM add noise at every step, so the same seed no longer yields unique output. Our observations about low-frequency components and our editing method don't apply to stochastic samplers. One workaround is to extend frequency-band editing to intermediate denoising steps, which fall into the combination of our method with SDEdit, as you mentioned in the review.
>
> ### **4. Flow-Matching Based Models**
> >How would this analysis change for flow-matching based models? I don't see a discussion of this, but I'm curious to know if similar patterns emerge.
>
> Flow-matching models extend diffusion models with deterministic sampling. We trained a flow-matching (FM) model with the OT scheduling function (or rectified flow) on AFHQ and compared it with pretrained EDM:
>
> | Network Pair | SSIM Paired | SSIM Random | MSE Paired | MSE Random |
> |--------------|-------------|-------------|------------|------------|
> | FM AFHQ vs EDM AFHQ | 0.7840 ± 0.1023 | 0.0581 ± 0.0454 | 0.0104 ± 0.0068 | 0.1168 ± 0.0447 |
> | FM AFHQ vs Patch PCA | 0.5116 ± 0.0862 | 0.0315 ± 0.0367 | 0.0394 ± 0.0149 | 0.1019 ± 0.0270 |
>
> This suggests flow-matching models (FM AFHQ) learn similar noise-to-image mappings as EDM AFHQ. This aligns with existing studies on noise-to-image map reproducibility across different architectures. For example, the work [24] shows different architectures (U-Nets, U-ViT, Consistency Trajectory Models) produce remarkably similar images given identical initial noise in their Fig. 1 and Fig. 11. Consequently, our observations and methods are directly applicable to flow-matching models. We will add these discussions to the main text.
>
> ### **5. [L114] Resampling the Frequency Band [n, 48]**
> >[L114] Can you please describe line 114 in more detail? i.e. "resampling the identified frequency bands [n,48] on each patch." An algorithm like algorithm 1 would be nice (can be included in the appendix).
>
> We decompose initial noise into 48 orthonormal Patch-PCA eigenvectors (3 channels × 4×4). To "resample" band [n,48], we set coefficients in range [n,48] to new i.i.d. Gaussian samples. Since the basis is orthonormal, overall noise remains Gaussian. We are happy to present this as an algorithm in the revision.
>
> ### **6. [Fig 1b] Row Description**
> >[Fig 1b] what do the three rows correspond to?
>
> Each set of three rows shows three independent examples (top to bottom) of images generated after perturbing initial noise in the specified frequency band (labeled above the first row).
>
> ### **7. [L132] Models Used**
> >[L132] Are different models are being used here? Are you using an ImageNet trained model A and a FFHQ trained model B and then generating images with the same seed?
>
> Yes. We use pixel-diffusion models separately trained on ImageNet, FFHQ, and AFHQ. For each model, we generate images with identical seeds to test whether similar global structure emerges across models trained on different datasets.
>
> ### **8. [Tab 1, Fig 5] LPIPS Metric Results**
> >[Tab 1, Fig 5] How do the differences measure in terms of the LPIPS metric?
>
> LPIPS results mirror SSIM trends. Same-seed images always receive lower scores (higher similarity) than random pairs. Differences are most pronounced in pixel-space models and smaller but evident for Patch PCA in latent space.
>
> **Pixel-space models:**
> | Network Pair | LPIPS (↓) Paired | LPIPS (↓) Random |
> |--------------|------------------|------------------|
> | ImageNet vs FFHQ | 0.454 ± 0.069 | 0.538 ± 0.064 |
> | ImageNet vs AFHQ | 0.486 ± 0.099 | 0.577 ± 0.093 |
> | ImageNet vs Patch PCA | 0.552 ± 0.101 | 0.626 ± 0.092 |
> | FFHQ vs AFHQ | 0.371 ± 0.093 | 0.449 ± 0.089 |
> | FFHQ vs Patch PCA | 0.542 ± 0.048 | 0.584 ± 0.044 |
> | AFHQ vs Patch PCA | 0.508 ± 0.067 | 0.560 ± 0.063 |
>
> **Latent-space models:**
> | Network Pair | LPIPS (↓) Paired | LPIPS (↓) Random |
> |--------------|------------------|------------------|
> | Stable Diffusion vs Patch PCA | 0.895 ± 0.059 | 0.951 ± 0.048 |
>
> ### **9. Patch-Size Sensitivity**
> >Was a patch size of 4x4 used for all experiments? How do results from the noise editing algorithm differ when using different patch sizes?
>
> Figure 1's frequency perturbation used 4×4 patches to divide 64×64 (or 32×32) images, preserving Gaussian properties. Image editing experiments used 5×5 patches for both pixel and latent space. We tested 3×3, 7×7, and 9×9 patches with similar qualitative results, though optimal eigenvector numbers and interpolation weights varied. Larger patches generally require more eigenvectors for comparable results.
>
> ---
>
> [24] Huijie Zhang, Jinfan Zhou, Yifu Lu, Minzhe Guo, Peng Wang, Liyue Shen, and Qing Qu. The emergence of reproducibility and consistency in diffusion models. ICML 2024.

---

> ### Comment · Reviewer_HRUH · 2025-08-08
> **Thanks**
>
> Thank you for responding to my questions. I maintain my recommendation for acceptance as I think this paper has insightful findings.
>
> > To "resample" band [n,48], we set coefficients in range [n,48] to new i.i.d. Gaussian samples.
>
> I realize this is late, so no worries if there is not enough time, but I would like to see the algorithm for this portion.
>
> > LPIPS Metric Results
>
> The difference in SSIM seems much higher than for LPIPS. Do you know why this might be the case?

---

> > ### Author Response · Authors · 2025-08-08
> >
> > Thank you for the supportive evaluation and thoughtful follow-up questions!
> >
> > ---
> >
> > ### 1 Resampling band $[n, 48]$ — algorithm
> >
> > *Input:* initial noise $z$ and cutoff index $n$ (with $1 \le n \le 48$).
> >
> > 1. Decompose $z$ onto the 48 patch-PCA eigenvector basis $\{v_i\}_{i=1}^{48}$ and record the coefficients $c_i = \langle z, v_i \rangle$.
> > 2. For every index $i \ge n$, replace $c_i$ with a fresh draw from standard Gaussian $\mathcal N(0,1)$; keep $c_i$ unchanged for $i < n$.
> > 3. Form the modified noise  $\tilde z = \sum_{i=1}^{48} c_i v_i .$
> >
> > Because the eigenvector basis is orthonormal and the original noise is i.i.d. Gaussian, each $c_i \sim \mathcal N(0,1)$, and the resampling maintains the Gaussian property. We will add a detailed algorithm to the updated Appendix.
> >
> > ---
> >
> > ### 2 SSIM versus LPIPS
> >
> > Indeed, SSIM shows a larger difference than LPIPS. LPIPS is a metric for perceptual which is more relevant to high-frequency content (see, for example, Fig. 1 in [Balestriero & LeCun 2024]), whereas SSIM places greater emphasis on low-frequency structure. The phenomenon that the difference in SSIM between identical noise pairs and random pairs is larger, while LPIPS shows a smaller change supports our finding that the initial noise primarily influences the low-frequency components of the final generation.
> >
> > [Balestriero & LeCun 2024] Balestriero, R. & LeCun, Y. How Learning by Reconstruction Produces Uninformative Features for Perception. ICML 2024.

---

### Official Review · Reviewer_t5kH · 2025-07-05

**Clarity:** 4
**Significance:** 3
**Originality:** 3
**Rating:** 4
**Confidence:** 3

**Summary:**

This paper explores the influence of initial noise seeds in diffusion models, focusing on how their low-frequency components encode universal compositional cues that govern the global structure of generated images. Through a detailed Patch-wise Principal Component Analysis (Patch PCA), the authors demonstrate that these structural signals persist consistently across different datasets and architectures, suggesting they are inherent properties of the generative process itself rather than artifacts of specific models or data.
Building on this insight, the authors propose a simple, interpretable, and training-free Patch PCA denoiser, which approximates the structural layout of generated images using only statistical properties of natural image patches. This denoiser is theoretically grounded under a local Gaussian patch distribution assumption and further validated through extensive experiments. Moreover, the paper introduces a zero-shot noise editing technique, which allows controllable manipulation of image composition by interpolating noise components within specific PCA frequency bands without model fine-tuning or additional training.

**Questions:**

I don't have other questions. Please refer to the weakness above.

**Ethical Concerns:**

["NO or VERY MINOR ethics concerns only"]

**Limitations:**

Yes

**Paper Formatting Concerns:**

No major formatting issues observed.

**Quality:**

3

**Strengths And Weaknesses:**

Strengths

●	The paper presents a compelling observation that the compositional structure of diffusion-generated images is predominantly determined by the low-frequency components of the initial noise. This is thoroughly validated via frequency band perturbation analysis, showing that perturbing low-frequency components significantly alters image structure, while high-frequency perturbations have negligible effect.

●	The compositional cues encoded in noise seeds are shown to generalize across architectures (e.g., pixel-based and latent diffusion models) and training datasets. This cross-domain consistency suggests a deeper universality in the noise-to-image mapping of diffusion models.

●	The idea of leveraging these structural cues for zero-shot image editing is novel and practical. The proposed editing approach operates solely on the noise level and maintains semantic coherence and structural layout using minimal assumptions. Qualitative examples show clear compositional transitions while preserving content.

●	The paper includes extensive visualizations, ablations, and quantitative analyses that strongly support the authors’ observation. The method is well-motivated both empirically and theoretically, and presented with clarity.

Weaknesses

●	While the Patch PCA denoiser is theoretically justified in pixel space under Gaussian patch assumptions, its extension to latent diffusion models is only evaluated empirically. Since the latent space is itself a learned low-dimensional representation, further theoretical or statistical analysis of PCA behavior in latent space would strengthen the argument for universality.

●	The core theoretical justification relies on the assumption that image patches follow a Gaussian distribution. Whether this assumption holds in latent space for latent diffusion models is unclear and not discussed.

●	Although the paper provides promising qualitative results for the zero-shot editing method, it lacks quantitative metrics (e.g., compositional similarity scores, human preference, or alignment accuracy) to objectively assess the success of compositional transfer. Moreover, claims that both structure and color can be selectively preserved or transferred through PCA manipulation are not fully substantiated by quantitative evidence.

---

> ### Author Rebuttal · Authors · 2025-07-30
>
> We thank the reviewer for the thoughtful and constructive feedback. We are encouraged that you found our central observations compelling, particularly the role of low-frequency components of initial noise in shaping compositional structure and the universality of this phenomenon across architectures and datasets. Below, we address the questions raised:
>
> ----
>
> ### 1. **Patch PCA in Latent Diffusion Models**
> >While the Patch PCA denoiser is theoretically justified in pixel space under Gaussian patch assumptions, its extension to latent diffusion models is only evaluated empirically. Since the latent space is itself a learned low-dimensional representation, further theoretical or statistical analysis of PCA behavior in latent space would strengthen the argument for universality.
>
> >The core theoretical justification relies on the assumption that image patches follow a Gaussian distribution. Whether this assumption holds in latent space for latent diffusion models is unclear and not discussed.
>
> We appreciate the reviewer's questions regarding Patch PCA in the latent space. Due to the nonlinearity of the VQ-VAE encoder, a Gaussian patch distribution may no longer remain statistically Gaussian in latent space. Extended theoretical analysis is an interesting direction, but it can be quite complicated. However, we observe that the encoder's dimensional reduction aligns remarkably well with PCA for **image patches**: the leading principal directions in pixel space strongly correspond to the principal directions in latent space—precisely the components our PCA-based editing method leverages.
>
> To support this, we conducted a subspace similarity analysis:
>
> * Step 1: Extracted 8p × 8p patches from natural images and computed the top k PCA directions in pixel space, where k ranges from 1 to 4p² (the dimensionality of the encoded latent patches).
> * Step 2: Encoded these PCA directions with the LDM encoder, yielding k vectors in latent space.
> * Step 3: Computed PCA directions directly on p × p latent patches from encoded images.
> * Step 4: Compared the two k-dimensional subspaces via mean cosine similarities of principal angles. The score ranges from 0 (orthogonal) to 1 (identical).
>
> Below are the results for 5×5 latent patches used in the paper:
>
> **First ten principal directions:**
>
> | k | 1 | 2 | 3 | 4 | 5 | 6 | 7 | 8 | 9 | 10 | Average |
> |-----|---|---|---|---|---|---|---|---|---|----|---------|
> | **Subspace Alignment ↑** | 0.9984 | 0.9042 | 0.9960 | 0.9969 | 0.8702 | 0.9926 | 0.9369 | 0.9410 | 0.9361 | 0.9380 | 0.9510 |
>
> **Coarser grid up to full latent-patch dimensionality (100):**
>
> | k | 10 | 20 | 30 | 40 | 50 | 60 | 70 | 80 | 90 | 100 | Average |
> |-----|----|----|----|----|----|----|----|----|----|----|---------|
> | **Subspace Alignment ↑** | 0.9380 | 0.8884 | 0.8739 | 0.9196 | 0.9468 | 0.9057 | 0.9409 | 0.9441 | 0.9530 | 0.9899 | 0.9300 |
>
> These results confirm our hypothesis with consistently high alignment values (>0.93 on average), supporting the validity of our patch PCA-based observations and methods in latent space. We will include this analysis in the revision.
>
> ### 2. **Quantitative Evaluation of Zero-Shot Editing**
>
> >Although the paper provides promising qualitative results for the zero-shot editing method, it lacks quantitative metrics (e.g., compositional similarity scores, human preference, or alignment accuracy) to objectively assess the success of compositional transfer. Moreover, claims that both structure and color can be selectively preserved or transferred through PCA manipulation are not fully substantiated by quantitative evidence.
>
> We agree that adding quantitative evaluation would further strengthen the experiments. To complement the qualitative results in Figure 7, we added quantitative evaluations across 256 random seeds using CIFAR-10. We use SSIM for structure alignment and Color Histogram Correlation for global color similarity. Color Histogram Correlation compares normalized RGB histograms using Bhattacharyya distance (range: 0 to 1, where 1 indicates identical distributions).
>
> | Metric | (a) No Edit | (b) Edit [1,9] | (c) Edit [1,9] \ {2,5} |
> |--------|-------------|----------------|------------------------|
> | SSIM vs. Ref ↑ | 0.0844 ± 0.0708 | 0.2521 ± 0.0665 | 0.2887 ± 0.0812 |
> | Color Hist. Corr. vs. Ref ↓ | 0.0402 ± 0.0860 | 0.2515 ± 0.0827 | 0.1110 ± 0.0987 |
> | Color Hist. Corr. vs. No Edit ↑ | N/A | 0.3809 ± 0.1137 | 0.5759 ± 0.1130 |
>
> Results show that editing low-frequency Patch PCA components significantly improves SSIM, confirming better compositional alignment. Excluding color channels (c) achieves better color preservation, as demonstrated by higher correlation with the no-edit case and lower correlation with the reference image. We will include these quantitative results in the revision.

---

### Comment · Area_Chair_qmxU · 2025-08-07

Dear Reviewers,

Thank you for your efforts. Please revisit the submission and check whether your earlier concerns have been adequately addressed in the author's response. If you haven't, please join the discussion actively before August 8 to let the authors know if they have resolved your (rebuttal) questions.

Best, AC

---

### Note · Authors · 2025-08-11

We sincerely thank all reviewers for their insightful feedback and constructive suggestions. We are encouraged by their consensus on our core contributions: a “strong and intuitive investigation” (azVL) providing an “elegant explanation” (1av2) of the correlation between initial noise and generated images, and the proposed zero-shot editing method is “novel and practical” (t5kH). Overall, our discovery of “universal compositional cues” is significant and “non-trivial and relevant to the broader diffusion community” (HRUH).

In response to feedback, we will make the following key additions in our revision:

1. **Quantitative validation:**  We will include quantitative analysis using SSIM and color histogram correlation to complement our qualitative visualizations, offering measurable evidence for compositional alignment and color preservation.

2. **Expanded universality:**  We will add the verification that our findings hold across deterministic samplers (EDM/VE/VP), flow-matching models, and transformer-based architectures (U-ViT), strengthening the universality claim.

3. **Demonstrated versatility:** We will add the experiments that extend the framework beyond low-frequency based composition (illuminations, major edges, and lines) to control texture (via high-frequency edits) and high-level composition (via an ImageNet object-counting task).

4. **Expanded theoretical and methodological support:** We will expand on the clarification of the optimality of Patch-PCA denoisers in nonlinear settings with patchwise Gaussian assumptions, discuss robustness when the data distribution is near patchwise Gaussian, and add subspace alignment analysis to connect it with dimension reduction in latent diffusion for image patches. We will also include a formal algorithm for frequency-band perturbation.

We are confident that these enhancements will address all reviewer concerns and further strengthen our paper.

---

### Decision · Program_Chairs · 2025-09-17

**Decision:**

Accept (poster)

**Comment:**

This paper investigates the influence of initial noise seeds on the final generation in diffusion models. Through patch-wise PCA analysis, the authors demonstrate that low-frequency components of the initial noise encode universal compositional cues that strongly affect the global structure of generated images.

Reviewers agree that this analysis is **non-trivial and insightful**, and the empirical evidence is clear. The rebuttal provided additional analyses, including extensions to high-frequency components, more quantitative evaluation, and validation across different architectures and samplers, which address most concerns raised by reviewers.

The rating current is two borderline accept, one accept, and one borderline reject.
Reviewer 1av2 maintained a borderline rejection after rebuttal, owing to limited evaluation of the proposed image editing algorithm and requesting broader experiments. However, Reviewer HRUH argued that the editing method is not the main contribution and that the core scientific finding stands on its own. AC agrees with this perspective. Additionally, one review’s final justification is missing, with an earlier rating of borderline accept.

Based on the strength of the main contribution and the quality of the rebuttal, AC recommends acceptance. The authors should incorporate key clarifications and additional results from the discussion into their revision.

-----------------

Lastly, AC would like to suggest that the authors discuss why this phenomenon occurs. Ideally, the initial noise should be pure Gaussian with a signal-to-noise ratio (SNR) close to zero. However, the model is still able to reveal information from the initial noise during inference. Could this originate from the training process? On the one hand, the forward process may not fully perturb the signal into pure noise [a]. On the other hand, since the denoising network is shared across timesteps, later steps expect higher-SNR inputs and may implicitly condition on residual information present in the initial noise. Will this knowledge be shared across timesteps and lead to the current phenomenon?

If it is caused by the SNR problem, will editing the high-frequency information be really possible? If we guarantee the SNR close to zero in training, and let the UNet for the initial denoising timestep not be shared, will the phenomenon still happen?

[a] Common Diffusion Noise Schedules and Sample Steps are Flawed